# From the Atlantic Coast to Lake Tanganyika: Gill-Infecting Flatworms of Freshwater Pellonuline Clupeid Fishes in West and Central Africa, with Description of Eleven New Species and Key to *Kapentagyrus* (Monogenea, Dactylogyridae)

**DOI:** 10.3390/ani11123578

**Published:** 2021-12-16

**Authors:** Maarten P. M. Vanhove, Raquel Hermans, Tom Artois, Nikol Kmentová

**Affiliations:** 1Centre for Environmental Sciences, Research Group Zoology: Biodiversity & Toxicology, Hasselt University, Agoralaan Gebouw D, 3590 Diepenbeek, Belgium; raquel.hermans@student.uhasselt.be (R.H.); tom.artois@uhasselt.be (T.A.); nikol.kmentova@uhasselt.be (N.K.); 2Department of Botany and Zoology, Faculty of Science, Masaryk University, Kotlářská 2, 611 37 Brno, Czech Republic

**Keywords:** Africa, biodiversity infrastructure, Clupeidae, Clupeiformes, Dactylogyridea, flatworm, historical collection, monogenea, Pellonulini, sardine

## Abstract

**Simple Summary:**

Sardines and other herring-like fishes (Clupeidae) are well-known, mostly from open seas, and globally commercially important. Their freshwater representatives receive less attention. Tropical Africa harbours over 20 species of the latter, classified under Pellonulini. These small river and lake fishes sustain locally important fisheries and are sometimes exported (inter)nationally. There is little research on them, let alone their parasites. An abundant parasite group of African freshwater clupeids is monogenean flatworms infecting their gills. Since the discoveries of the first (1969) and second species (1973) systematics of these monogeneans was ignored until 2018, when they were classified under the new genus *Kapentagyrus* with three species from three pellonuline species. Here, we inspected the gills of 12 West and Central African pellonulines, 10 from which there were no known parasites. We discovered and described 11 new species of *Kapentagyrus*. They look highly similar; distinguishing them requires measuring parts of their attachment organ. This study more than quadruples the known species richness of *Kapentagyrus*, and almost quadruples the number of pellonuline species of which monogeneans are known. Monogeneans are suitable tags for the lifestyle and history of their hosts. Therefore, parasitological knowledge on these poorly studied fishes will contribute to understanding data-poor African fisheries.

**Abstract:**

Unlike their marine counterparts, tropical freshwater clupeids receive little scientific attention. However, they sustain important fisheries that may be of (inter)national commercial interest. Africa harbours over 20 freshwater clupeid species within Pellonulini. Recent research suggests their most abundant parasites are gill-infecting monogenean flatworms within *Kapentagyrus*. After inspecting specimens of 12 freshwater clupeids from West and Central Africa, mainly sourced in biodiversity collections, we propose 11 new species of *Kapentagyrus,* which we describe using their haptoral and genital morphology. Because of their high morphological similarity, species delineation relies mostly on the morphometrics of anchors and hooks. Specifically, earlier, molecular taxonomic work indicated that the proportion between the length of the anchor roots, and between the hook and anchor length, is diagnostic. On average, about one species of *Kapentagyrus* exists per pellonuline species, although *Pellonula leonensis* harbours four species and *Microthrissa congica* two, while *Microthrissa moeruensis* and *Potamothrissa acutirostris* share a gill monogenean species. This study more than quadruples the number of known species of *Kapentagyrus*, also almost quadrupling the number of pellonuline species of which monogeneans are known. Since members of *Kapentagyrus* are informative about their hosts’ ecology, evolutionary history, and introduction routes, this enables a parasitological perspective on several data-poor African fisheries.

## 1. Introduction

The bulk of pelagic marine fish catches, and, hence, the largest fisheries on the planet, are based on sardines and anchovies, representatives of Clupeiformes [1]. Less species-rich and less well-studied are the freshwater clupeiformes, such as the members of Pellonulini. This is a tribe of dorosomatine clupeids restricted to Afrotropical freshwaters, containing ca. 22 species [2]. Representatives of Pellonulini constitute the major part of commercial fish catches in, e.g., Lakes Tanganyika, Mweru, Kivu, and Kariba [3], and also contribute to important subsistence fisheries elsewhere in Africa [4]. Moreover, given the rising importance of freshwater fisheries as a protein source in Africa [5], we expect their economic and ecological role to increase. Their proportion in African freshwater fish catches has increased during the second half of the 20th century, towards almost half of the total tonnage over a range of African lakes (not including Lake Victoria). Moreover, clupeids are among the fishes that can quickly and abundantly colonise the newly formed pelagic habitat after damming African rivers [3].

While various timings were proposed, there seems to be a consensus that pellonuline clupeids diversified in African inland waters after a single marine-to-freshwater transition [2,6]. In this context, it is particularly interesting to inventory host–parasite interactions as parasites may be a useful source of information on the history of freshwater colonisation by their fish hosts (neotropical examples: [7,8]). Parasite species (either monogeneans or crustaceans) have only been reported in four species of Pellonulini [9,10]. It is, therefore, not only hard to reconstruct marine-to-freshwater introduction pathways but also to estimate to what extent parasites can devastate these clupeid stocks. Other small pelagic fish are known to be affected by parasite infection, such as the cyprinid *Rastrineobola argentea* (Pellegrin, 1904) in Lake Victoria by the diphyllobothriid tapeworm *Ligula intestinalis* (Linnaeus, 1758) [11]. Another example of the management relevance of fish parasites is their potential to help elucidate the origin and pathway of anthropogenic fish translocations. This was demonstrated by, e.g., Mombaerts et al. [12] for monogenean flatworms infecting Ponto-Caspian gobies invasive in Belgium. Fish–parasite systems in coastal regions may also provide insight into diversification processes of both actors as marine transgressions and regressions can cause isolation, reconnection, and demographic changes in hosts as well as parasites [13]. 

Monogenean flatworms fulfil several criteria listed by Catalano et al. [14] for parasites to make good biological tags: they are relatively easy to collect because they are often ectoparasitic, and their one-host lifecycle and often low pathogenicity avoid confounding factors in interpreting host–parasite relationships. As the pellonuline *Limnothrissa miodon* (Boulenger, 1906) has been introduced to several African water bodies [4], the monogenean fauna of pellonuline sardines also merits exploration from the point-of-view of invasion biology. Kmentová et al. [15] provide a compelling case of parasites as tags for the introduction of freshwater sardines as the occurrence and morphology of representatives of *Kapentagyrus* Kmentová, Gelnar & Vanhove, 2018 (Dactylogyridae) confirm reports on the methods used for translocating *L. miodon*.

Currently, the only African freshwater clupeids from which monogenean parasites are known are *Limnothrissa miodon*, *Pellonula leonensis* Boulenger, 1916, and *Stolothrissa tanganicae* Regan, 1917. They are infected by one (*P. leonensis*, *S. tanganicae*) or two (*L. miodon*) species of *Kapentagyrus* [9]. This genus is unrelated to the monogenean ectoparasites of marine clupeids [9,16], suggesting that this parasite fauna was lost and replaced upon the clupeids’ incursion into African freshwater. While this is the expected scenario for such marine–freshwater transitions [17], some marine monogenean lineages have persisted after colonisation of freshwater, e.g., *Dolicirroplectanum* Kmentová, Gelnar & Vanhove, 2020 (Diplectanidae) on lates perches [18], and *Euryhaliotrema* Kritsky & Boeger, 2002 (Dactylogyridae) on sciaenids [8]. One can, therefore, wonder whether all African pellonuline clupeids harbour representatives of *Kapentagyrus*, or whether there are also species among them that host typically marine monogeneans.

In an effort to increase understanding of the host distribution of species of *Kapentagyrus* throughout pellonuline clupeids, we scrutinise the gills of a number of West and Central African pellonulines for monogenean parasites. Given the monogenean richness described from other African pellonulines, we expect to find at least one species of *Kapentagyrus* on each clupeid species. As the tribe Pellonulini is relatively well-represented in museum collections, our research question and hypothesis can be approached in a logistically efficient and environmentally friendly way through a collection-based parasitological survey with an extensive host and geographical coverage. The recently increased attention for such a museum-based approach gave rise to various helminthological discoveries [19]. Collection-based work offers a great deal of untapped potential in terms of baseline reconstruction in dactylogyrid monogeneans as they can be morphologically identified to the species level when retrieved from historical fish specimens [20].

## 2. Materials and Methods

### 2.1. Collection and Availability of Specimens 

The right-side gill arches of host specimens (typically formaldehyde-fixed and preserved in denatured ethanol) from the ichthyology collections of the Royal Museum for Central Africa (RMCA) (Tervuren, Belgium) and the Natural History Museum (London, UK) (Figure 1; Table 1) were dissected and inspected for monogenean parasites using mounted needles. In addition, fresh specimens were acquired commercially (from a fishmonger) from Zambian Lake Itezhi-Tezhi, of which both gill chambers were screened for monogeneans. Part of the monogenean specimens were partly digested using a 9:1 ratio of 10 mg/μL proteinase *K* and TNES buffer. The digestion was stopped by washing specimens in distilled H_2_O. Monogeneans were placed on a slide in a drop of water that was subsequently replaced by Hoyer’s medium and covered with a cover slip that was sealed with nail polish. Part of the retrieved specimens was cut in half to produce hologenophores for future molecular characterisation. Type material was deposited in the invertebrate collection of the RMCA (RMCA_VERMES_43430-459), the collection of the research group Zoology: Biodiversity and Toxicology of Hasselt University (Diepenbeek, Belgium) (HU 779-820), the Finnish Museum of Natural History (Helsinki, Finland) (MZH https://id.luomus.fi/KV.670-https://id.luomus.fi/KV.685), and the Iziko South African Museum (Cape Town, South Africa) (SAMC-A094487-504). Parasite type specimens are linked to the host specimens here designated as symbiotypes [21] and symbioparatypes [22]. To comply with the regulations set out in article 8.5 of the amended 2012 version of the International Code of Zoological Nomenclature (ICZN) [23], details of the species have been submitted to ZooBank. The Life Science Identifier (LSID) of the article is urn:lsid:zoobank.org:pub:C8EB57D2-8C77-4298-ACF1-24CE2D8A409D.

### 2.2. Microscopy and Morphometrics

Parasites were observed and measured under Olympus BX51 and Leica DM2500LED microscopes at a magnification of ×1000 (objective ×100 immersion, ocular ×10). Differential diagnosis and species description focus on the morphology of the flatworms’ haptoral and genital hard parts, as in Kmentová et al. [9], integrating morphological and genetic data, proving their utility in species delineation within *Kapentagyrus*. These structures were measured using the same metrics that Kmentová et al. [9] used for the original description of *Kapentagyrus* (Figure 2). To evaluate the level of inter- versus intraspecific variability of collected monogenean specimens, scaled morphometric data of the haptoral region were analysed by principal component analysis (PCA) in the R package ade4 v1.7.18 [24] and non-metric multidimensional scaling (NMDS) in the R package vegan v2.5.7 [25] using autotransformation (variables and specimens with more than 50% of missing data were omitted from the analyses). To allow a comparison with previously characterised Tanganyika species, we added morphometric data of *K. limnotrissae* (Paperna, 1973) and *K. tanganicanus* Kmentová, Gelnar & Vanhove, 2018 of specimens (Figure 1) prepared using the same protocol as described above ([26], Mendeley Data, V2, doi:10.17632/jvz2m6y9nt.2) (to control for the effect of different fixation methods being used [26,27]) to the multivariate statistical analyses. To recognise and delineate species new to science, we adhere to the phylogenetic species concept. It proposes that species are reproductively isolated groups of natural populations, originating through a speciation event and ending with the next speciation or vanishing through extinction [28]. In practice, we defined a group of specimens as belonging to a new species of *Kapentagyrus* when we found them to consistently differ from another group of congeners in minimally one attribute [29]. Results of the PCA and NDMS were visualised with the packages ggplot2 v3.3.5 [30] and factoextra v1.0.7 [31]. The map depicting sampling localities was created using qgis v3.16 (QGIS Development Team 2021. QGIS Geographic Information System. Open Source Geospatial Foundation Project. http://qgis.osgeo.org, accessed on 23 October 2021).

## 3. Results

### 3.1. Negative Results

*Limnothrissa miodon* (n = 24) from the introduced population in Lake Itezhi-Tezhi, *Pellonula vorax* Günther, 1868 from Lake Nguene (n = 3), *Potamothrissa acutirostris* (Boulenger, 1899) from Ankoro (n = 2), Tshopo River (n = 2), and Bosabangi (n = 1), and *Potamothrissa obtusirostris* (Boulenger, 1909) from Lualaba River (n = 1) and Oso River (n = 1) were screened for gill monogeneans without success. 

### 3.2. Multivariate Statistics

Morphological variation of *Kapentagyrus* spp. was visualized based on a PCA performed on 20 standardised haptoral morphometric parameters. The first PCA included morphometric data on haptoral structures of specimens of all host species analysed in this study. The seventh pair of hooks was excluded due to the large number of missing data. The first two PC axes explained 41.7% and 20.8% of the variation, respectively (Figure 3a). Separation of species infecting members of *Odaxothrissa* Boulenger, 1899 and *Kapentagyrus verbisti* n. sp. ex *P. leonensis* from Lambaréné (Gabon) was visible along the first PC axis mainly driven by parameters from both dorsal and ventral anchors. Clustering with rather a continuous gradient along both PC axes among the rest of the species of *Kapentagyrus* suggests close morphological similarity among the species within *Kapentagyrus* with the first PC axis explaining 26.2% and the second 22.4% of the variation present in the dataset (Figure 3b). In the NMDS plots (Figure 4), clusters and the length and direction of the vectors of the most important parameters in PCA are similar. While these parameters clearly contribute a great deal to differentiating species of *Kapentagyrus*, these metrics alone do not allow complete separation between species. Likewise, the length of the copulatory tube (Figure 5a) and accessory piece (AP) (Figure 5b) of the male copulatory organ (MCO) substantially overlap between various species of *Kapentagyrus*.

### 3.3. Taxonomic Account

All collected flatworms belong to *Kapentagyrus* as they show all diagnostic features of this genus: dactylogyrid monogeneans with two pairs of haptoral anchors, all of which carry two well-developed roots, two simple, V-shaped haptoral transversal bars without auricles, and an MCO with a simple copulatory tube intertwined by a similar sized AP. Just like in all hitherto described species of *Kapentagyrus*, a sclerotised vagina was not observed in the species in the current study. In total, 11 new species of *Kapentagyrus* are described below. In what follows, for each species, we will only mention the diagnostic features of haptor and MCO given their utility in species delineation within *Kapentagyrus* [9]. We will also discuss for each species how it differs from all other species of the genus. Measurements are given in Table 2, Table 3 and Table 4. 

Family: Dactylogyridae Yamaguti, 1963

Genus: *Kapentagyrus* Kmentová, Gelnar & Vanhove, 2018

#### 3.3.1. Description of *Kapentagyrus voreli* n. sp.

*Kapentagyrus voreli* n. sp. (Figure 6; Table 2)

Type material: holotype (RMCA_VERMES_43431) and eight paratypes (RMCA_VERMES_43431, HU 779, 780, 803, SAMC A094487-8, MZH https://id.luomus.fi/KV.670, https://id.luomus.fi/KV.671)

Symbiotype: MRAC A1-070-P-0266

Symbioparatypes: MRAC A0-048-P-1252, A0-048-P-1255-56, A0-048-P-1260

Type host: *Odaxothrissa ansorgii* Boulenger, 1910 (Teleostei: Clupeidae)

Type locality: Noya River near Aboumé, Gabon

Other locality: Lake Nguene (Ogooué Basin), Gabon

ZooBank registration: The Life Science Identifier (LSID) for *Kapentagyrus voreli* Vanhove, Hermans, Artois & Kmentová n. sp. is urn:lsid:zoobank.org:act:7C165764-9233-4B9E-ACDF-86D7D9EB7E1C.

Material examined: four whole-mounted and five partly mounted specimens

Infection site: gill filaments

Infection parameters: Noya River: one out of two hosts infected with one worm (infection intensity = 1); Lake Nguene: four out of ten hosts infected with one to three worms (infection intensity = 2) 

Etymology: the species epithet honours biologist Jiří Vorel (Czech Republic) for his passion in furthering bio-informatic and genomic research on non-model parasitic flatworms, including members of *Kapentagyrus*.

Diagnosis: a pair of dorsal and ventral anchors with long blade; inner root of anchors much broader and about three times longer than outer root, with wide incision between them. Dorsal and ventral bars enlarged in the middle and towards the extremities, extremities blunt. Hooks seven pairs, similar in length. Male copulatory organ with small, irregularly shaped heel; AP longer than copulatory tube, winding or looping once around the copulatory tube from the right side. 

Discussion: compared to the three species of *Kapentagyrus* hitherto described, *K. voreli* n. sp. clearly differs in the size of the dorsal anchors. While their total length is below 33 µm in the two Tanganyika species, *K. limnotrissae* and *K. tanganicanus*, they measure between 35 µm and 39 µm in *K. voreli* n. sp. Conversely, the dorsal anchors are at least 40 µm long in *K. pellonulae* (Paperna, 1969). The same goes for the ventral anchors, which are 35 to 37 µm long in *K. voreli* n. sp., considerably longer (50 µm) in *K. pellonulae*, and below 32 µm in *K. limnotrissae*. There may be some overlap in ventral anchor length between *K. voreli* n. sp. and *K. tanganicanus* (where Kmentová et al. [9] also report a maximum ventral anchor length of 37 µm, but a lower average length of 32 µm or 19 µm depending on the host species). However, there are also considerable shape differences in the anchors, with those of *K. voreli* n. sp. exhibiting a thinner anchor blade than the Tanganyika species and a more slender general appearance of the anchors, especially the dorsal one.

#### 3.3.2. Description of *Kapentagyrus marispastoris* n. sp.

*Kapentagyrus marispastoris* n. sp. (Figure 7; Table 2)

Type material: holotype (RMCA_VERMES_43432)

Symbiotype: MRAC 1973.007.P.0019(1)

Type host: *Odaxothrissa mento* (Regan, 1917) (Teleostei: Clupeidae)

Type locality: Lake Volta at Yeji, Ghana

ZooBank registration: The Life Science Identifier (LSID) for *Kapentagyrus marispastoris* Vanhove, Hermans, Artois & Kmentová n. sp. is urn:lsid:zoobank.org:act:31A02C62-9C93-4FFA-89D8-C086A3C730B5.

Material examined: one whole-mounted specimen

Infection site: gill filaments

Infection parameters: one out of three hosts infected with one worm (infection intensity = 1)

Etymology: the species epithet is a genitive derived from the Latin words “mare” (“sea”, gen.: maris) and “pastor” (shepherd, gen.: pastoris) in honour of the international non-profit organization Sea Shepherd for its efforts towards marine conservation in general and against illegal, unreported, and unregulated fishing in Africa in particular. 

Diagnosis: a pair of dorsal and ventral anchors with inner roots broader than and about twice as long as outer roots; narrow incision between them. Dorsal and ventral bars enlarged in the middle and towards the extremities, extremities blunt, bar branches of similar length. Hooks seven pairs, similar in length. Male copulatory organ with AP longer than copulatory tube, winding or looping once around the copulatory tube from the right side, reaching its distal end; basal bulb of copulatory tube embedded in irregular heel-like structure. 

Discussion: the length and slender appearance of the anchors of *K. marispastoris* n. sp., and their slender blades, are reminiscent of its congener on the gills of the congeneric host *O. ansorgii*, *K. voreli* n. sp. However, the inner and outer root length of the dorsal anchor of *K. marispastoris* n. sp. (22 µm and 9 µm) exceed those of *K. voreli* n. sp. (15 to 18 µm, and 4 to 6 µm).

#### 3.3.3. Description of *Kapentagyrus sefcae* n. sp.

*Kapentagyrus sefcae* n. sp. (Figure 8; Table 2)

Type material: holotype (RMCA_VERMES_43434) and 15 paratypes (RMCA_VERMES_43433, 35, 36, HU 781-6, 805, SAMC A094489-92, MZH https://id.luomus.fi/KV.672)

Symbiotype: MRAC 1977.042.P.0003

Symbioparatype: MRAC 1977.042.P.0004

Type host: *Odaxothrissa losera* Boulenger, 1899 (Teleostei: Clupeidae)

Type locality: Pool Malebo, Congo River, Democratic Republic of the Congo

ZooBank registration: The Life Science Identifier (LSID) for *Kapentagyrus sefcae* Vanhove, Hermans, Artois & Kmentová n. sp. is urn:lsid:zoobank.org:act:A5AF784B-6202-4013-A1B7-4F36BA26F1AE.

Material examined: eight whole-mounted and ten partly mounted specimens

Infection site: gill filaments

Infection parameters: two out of two hosts infected with three or sixteen worms (infection intensity = 9.5)

Etymology: the species epithet honours biologist Kristina Sefc (Austria), professor of evolutionary biology at the University of Graz, for her contributions to research and supervision on African cichlids.

Diagnosis: a pair of dorsal and ventral anchors with long blade; inner root of anchors much broader than outer root, with narrow, shallow incision between them. Dorsal and ventral bars enlarged in the middle, extremities blunt, with dorsal bar branches slightly longer. Hooks seven pairs, similar in length. Male copulatory organ with small, irregular heel at proximal end of basal bulb, AP longer than copulatory tube, winding once around the copulatory tube from the left side, reaching its distal end. 

Discussion: the long anchor blades remind of the congeners infecting members of the same host genus, namely *K. voreli* n. sp. and *K. marispastoris* n. sp. *Kapentagyrus sefcae* n. sp. can be distinguished from both by (1) a total length of the ventral anchor of more than 40 µm compared to maximum 38 µm in *K. voreli* n. sp. and *K. marispastoris* n. sp., (2) the inner root length of the ventral anchor of at least 23 µm, whereas this is maximally 19 µm in *K. voreli* n. sp. and *K. marispastoris* n. sp., and (3) a narrow and shallow incision between the inner and outer root of the ventral anchors, which is wider in *K. voreli* n. sp.

#### 3.3.4. Description of *Kapentagyrus parisellei* n. sp.

*Kapentagyrus parisellei* n. sp. (Figure 9; Table 2)

Type material: holotype (RMCA_VERMES_43457) and 12 paratypes (RMCA_VERMES_43458-9, HU 800-2, 817-8, SAMC A094504-6, MZH https://id.luomus.fi/KV.684, https://id.luomus.fi/KV.685)

Symbiotype: MRAC P.93619

Symbioparatypes: MRAC P.93614, P.93617, P.93619, P.100646-49, P.100652, P.100655

Type host: *Nannothrissa parva* (Regan, 1917) (Teleostei, Clupeidae)

Type locality: Lake Tumba, Democratic Republic of the Congo

Other localities: tributary of Lake Tumba, Democratic Republic of the Congo

ZooBank registration: The Life Science Identifier (LSID) for *Kapentagyrus parisellei* Vanhove, Hermans, Artois & Kmentová n. sp. is urn:lsid:zoobank.org:act:313E0774-199D-4D1D-AC8A-04EA45AC50D5.

Material examined: nine whole-mounted and four partly mounted specimens

Infection site: gill filaments

Infection parameters: Lake Tumba: two out of nine hosts infected with one or two worms each (infection intensity = 1.5); tributary of Lake Tumba: six out of ten hosts infected with one or two worms each; Mbandaka (then known as Coquilhatville): none of two hosts infected

Etymology: the species epithet honours biologist Antoine Pariselle (France/Morocco), senior researcher at the Institut de Recherche pour le Développement, one of the most productive specialists of African monogeneans and a champion of capacity development in fish parasitology in the Global South and North. 

Diagnosis: a pair of dorsal and ventral anchors, asymmetrical in shape: ventral anchor inner root ca. thrice as long as its outer root, length difference between inner and outer root much smaller in dorsal anchor. Short and compact anchor blade. Dorsal and ventral bars V-shaped, enlarged in the middle, with branches of comparable length but on average slightly longer in dorsal bar. Hooks seven pairs. Enlarged part of the copulatory tube pear-shaped, starting in the bulb and tapering until almost halfway the copulatory tube. Copulatory tube opens terminally. Accessory piece circles around the copulatory tube in a clear loop from the right side, then follows its course without always reaching its distal end. 

Discussion: *Kapentagyrus parisellei* n. sp. differs from *K. limnotrissae* and *K. tanganicanus* infecting clupeids in Lake Tanganyika (*L. miodon* and *S. tanganicae*) by its copulatory tube that is enlarged far beyond the basal bulb, to the extent the copulatory tube takes a pear shape until about halfway to its axial length. In the congeners from Lake Tanganyika, the copulatory tube narrows immediately distally from the basal bulb. In contrast to its congeners on the gills of members of *Odaxothrissa*, the dorsal anchors of *K. parisellei* n. sp. have a compact, more stubby look as they lack the long and slender blades. *Kapentagyrus parisellei* n. sp. is unique among its congeners in the combination of a first hook pair longer than the inner root length of the dorsal anchor and a length to notch of the ventral anchor of less than 19 µm.

#### 3.3.5. Description of *Kapentagyrus hugei* n. sp.

*Kapentagyrus hugei* n. sp. (Figure 10; Table 3)

Type material: holotype (RMCA_VERMES_43445) and 32 paratypes (RMCA_VERMES_43441-4,46-7, HU 789-5, 806-11, 819-20, SAMC A094493-8, MZH https://id.luomus.fi/KV.674, https://id.luomus.fi/KV.675, https://id.luomus.fi/KV.676, https://id.luomus.fi/KV.677, https://id.luomus.fi/KV.678, https://id.luomus.fi/KV.679)

Symbiotype: MRAC 1973.005.P.0362

Symbioparatypes: MRAC 1973.005.P.0356, MRAC 1973.005.P.0358, MRAC 1973.005.P.0361, MRAC 1973.005.P.0363, MRAC 1973.005.P.0365-9, MRAC 1973.005.P.0371-3

Type host: *Pellonula leonensis* Boulenger, 1916 (Teleostei, Clupeidae)

Type locality: Lake Volta, Ghana

ZooBank registration: The Life Science Identifier (LSID) for *Kapentagyrus hugei* Vanhove, Hermans, Artois & Kmentová n. sp. is urn:lsid:zoobank.org:act:4D1130CA-4295-4971-81EE-CEC925BE16E6.

Material examined: 22 whole-mounted and 11 partly mounted specimens

Infection site: gill filaments

Infection parameters: 13 out of 19 hosts infected with one to nine worms each.

Etymology: the species epithet honours bioscience engineer Jean Hugé (Belgium/the Netherlands) for his contributions to sustainability science and conservation science in the Global South.

Diagnosis: ventral and dorsal anchors similar in size, asymmetrical in root lengths with inner root two to three times as long as outer one; ventral and dorsal bars V-shaped and of similar length, both strongly tapering towards the distal ends of the branches; ventral bar strongly thickened in the middle; hooks seven pairs of similar length; simple MCO with thin AP looping around the copulatory tube from the right, copulatory tube beginning in triangular basal bulb that smoothly transitions into the rest of the tube. 

Discussion: *Kapentagyrus hugei* n. sp. differs from *K. tanganicanus* and *K. limnotrissae* by the larger dorsal anchors: total average length 28 µm in *K. hugei* n. sp. versus 20 µm in *K. limnotrissae* and 25 µm or 22 µm in *K. tanganicanus* depending on the host species. The difference with its congener infecting *N. parva* lies in the total length of the dorsal anchor (26 to 31 µm in *K. hugei* n. sp. versus 18 to 22 µm in *K. parisellei* n. sp.). This parasite of *P. leonensis* clearly differs from the monogeneans infecting members of *Odaxothrissa* as it lacks their characteristic elongated, slender dorsal anchor shape.

#### 3.3.6. Description of *Kapentagyrus hahni* n. sp.

*Kapentagyrus hahni* n. sp. (Figure 11; Table 3)

Type material: holotype (RMCA_VERMES_43437) and two paratypes (RMCA_VERMES_43438, HU 787)

Symbiotype: NHMUK 1964.10.12.27

Symbioparatypes: NHMUK 1964.10.12.24, NHMUK 1964.10.12.29

Type host: *Pellonula leonensis* Boulenger, 1916 (Teleostei, Clupeidae)

Type locality: Agorkpo Creek, Lower Volta Basin, Ghana

ZooBank registration: The Life Science Identifier (LSID) for *Kapentagyrus hahni* Vanhove, Hermans, Artois & Kmentová n. sp. is urn:lsid:zoobank.org:act:6B65666A-A37F-44E6-ABA5-B56A8881D260.

Material examined: three whole-mounted specimens

Infection site: gill filaments

Infection parameters: three out of seven hosts infected with one or two worms each (infection intensity: 1.3).

Etymology: the species epithet honours biologist Christoph Hahn (Austria) for his seminal contributions to genomics in monogeneans and other non-model organisms.

Diagnosis: ventral anchors highly asymmetrical in root length; roots thin with sharp proximal ends; ventral anchor appears compact, “flattened” along the anterior-posterior axis because of the short length to notch; ventral and dorsal bars V-shaped; ventral bar thickest in the middle and tapering towards the ends; dorsal bar of irregular thickness along its branch lengths; hooks seven pairs of similar length; inconspicuous, simple MCO with AP, intertwined around the copulatory tube from the right side, copulatory tube beginning in enlarged basal bulb.

Discussion: the shape of the anchors and bars of *K. hahni* n. sp. are somewhat reminiscent of the description of *K. pellonulae,* which infects the same host [32]. However, the incisions between the anchor roots are wider [32]. There are also considerable size differences between both species, e.g., in total length of the dorsal (max. 30 µm in *K. hahni* n. sp. versus min. 40 µm in *K. pellonulae*) and ventral anchor (max. 29 µm versus 50 µm). Compared to its congener *K. hugei* n. sp. infecting the same host species also in the Volta Basin, *K. hahni* n. sp. has more slender anchor roots, and a shorter length to notch in the ventral anchor (< 19 µm in *K. hahni* n. sp. versus >19 µm in *K. hugei* n. sp.). Moreover, the angle between the inner root and the inner side of the blade of the ventral anchor is wider in *K. hahni* n. sp., while it is sharper in *K. hugei* n. sp. Moreover, the outer side of the blade aligns more continuously with the outer root of the ventral anchor in *K. hahni* n. sp. than in *K. hugei* n. sp. and all other congeners. These characters related to the anchor shape were also used by Pariselle et al. [33] to distinguish members of *Cichlidogyrus* Paperna, 1960 infecting the cichlid *Interochromis loocki* (Poll, 1949) from their congeners infecting closely related cichlids.

#### 3.3.7. Description of *Kapentagyrus verbisti* n. sp.

*Kapentagyrus verbisti* n. sp. (Figure 12; Table 3)

Type material: holotype (RMCA_VERMES_43435) and three paratypes (RMCA_VERMES_43434, HU 788, MZH https://id.luomus.fi/KV.673)

Symbiotype: MRAC 2000.048.P.1121

Symbioparatypes: MRAC 2000.048.P.1120

Type host: *Pellonula leonensis* Boulenger, 1916 (Teleostei, Clupeidae)

Type locality: Lambaréné, Gabon

ZooBank registration: The Life Science Identifier (LSID) for *Kapentagyrus verbisti* Vanhove, Hermans, Artois & Kmentová n. sp. is urn:lsid:zoobank.org:act:E0616EDC-E497-4691-9BB8-902AD8386445.

Material examined: two whole-mounted and two partly mounted specimens

Infection site: gill filaments

Infection parameters: two out of six hosts infected with two to three worms (infection intensity 2.5).

Etymology: the species epithet honours bioscience engineer Bruno Verbist (Belgium) for his contributions to sustainability science and agroforestry research in tropical Africa and Asia.

Diagnosis: dorsal and anchors with broad roots and narrow incision. Ventral bar yoke-shaped, tapering towards the distal end of its branches, blunt end of branches. Dorsal bar widens towards the blunt distal ends of branches. Copulatory tube of MCO with well-developed heel, funnel-shaped basal bulb. Accessory piece strongly curved, winds around copulatory tube.

Discussion: *Kapentagyrus verbisti* n. sp. differs from all other congeners in the large curves (oriented almost perpendicularly) in the accessory piece as it winds around the copulatory tube.

#### 3.3.8. Description of *Kapentagyrus chochamandai* n. sp.

*Kapentagyrus chochamandai* n. sp. (Figure 13 and Figure 14; Table 4)

Type material: holotype (RMCA_VERMES_43448) and six paratypes (RMCA_VERMES_43449, HU 796, 812-3, SAMC A094499-500, MZH https://id.luomus.fi/KV.680)

Symbiotype: MRAC 1993.145-0620 

Symbioparatypes: MRAC 1993.145-0611, MRAC 1993.145-0614, MRAC 1994.019.P.2064, MRAC 1994.019.P.2068, MRAC 1994.019.P.2080, NHM 1920.5.26.3-12

Type host: *Microthrissa moeruensis* (Poll, 1948) (Teleostei, Clupeidae)

Other host: *Potamothrissa acutirostris* (Boulenger, 1899) (Teleostei, Clupeidae)

Type locality: Lake Mweru, Zambia

Other locality: Kashilu, Zambia; Kilwa, Lake Mweru, Democratic Republic of the Congo

ZooBank registration: The Life Science Identifier (LSID) for *Kapentagyrus chochamandai* Vanhove, Hermans, Artois & Kmentová n. sp. is urn:lsid:zoobank.org:act:6E37DF9D-3676-4C55-80E6-62DA1BCD2422.

Material examined: eight whole-mounted specimens

Infection site: gill filaments

Infection parameters: Lake Mweru, on type host: three out of ten hosts infected with one worm each (average infection intensity = 1); Kilwa, Lake Mweru, on *P. acutirostris*: one out of ten hosts infected with one worm (infection intensity = 1); Kashilu: two out of five hosts infected with one worm (average infection intensity = 1) 

Etymology: the species epithet honours bioscience engineer Auguste Chocha Manda (Democratic Republic of the Congo), professor at the Université de Lubumbashi, in recognition of his efforts for Congolese ichthyological, ichthyoparasitological, and aquacultural research, including in the basin where the type locality of the species is.

Diagnosis: a pair of dorsal and ventral anchors each with inner root two to three times as long as outer root, short and compact anchor blade. Dorsal and ventral bars V-shaped, enlarged in the middle, with branches of similar length. Hooks seven pairs, all similar in length. Male copulatory organ with copulatory tube that starts in an enlarged bulb, tapers distally and opens terminally, and AP winding once around the copulatory tube from the left side, longer than copulatory tube and reaching its distal end. 

Discussion: *Kapentagyrus chochamandai* n. sp. differs from its congeners infecting representatives of clupeids in Lake Tanganyika (*L. miodon* and *S. tanganicae*) by the shorter average length of the point of the dorsal anchor: 6 µm in *K. chochamandai* n. sp. versus 8 µm in *K. limnotrissae* and *K. tanganicanus* (data from [9]). It can be distinguished from the species of *Kapentagyrus* infecting members of *Odaxothrissa* by the more stubby, less elongated haptoral anchors. The difference between *K. chochamandai* n. sp. and *Kapentagyrus parisellei* n. sp. lies in the orientation of the AP loop around the copulatory tube (from the left versus from the right in *K. parisellei* n. sp.), as well as the length to notch of the ventral anchor (at least 19 µm versus less than 19 µm in *K. parisellei* n. sp.). The difference between *K. chochamandai* n. sp. and its congeners infecting *P. leonensis* lies in: (1) the more elongated appearance of the dorsal anchors of *K. hugei* n. sp. and *K. verbisti* n. sp., with a length to notch of 21 to 25 µm and 26 to 28 µm, respectively, versus 17 to 19 µm in *K. chochamandai* n. sp.; (2) the smooth transition between the outer root and blade of the ventral anchor in *K. hahni* n. sp., where a clear indentation is present in *K. chochamandai* n. sp.; and (3) the larger haptoral anchors in *K. pellonulae*: minimally 40 µm, and 50 µm, for dorsal and ventral anchor, respectively, versus maximally 26 µm and 34 µm in *K. chochamandai* n. sp.

#### 3.3.9. Description of *Kapentagyrus bisthoveni* n. sp.

*Kapentagyrus bisthoveni* n. sp. (Figure 15; Table 4)

Type material: holotype (RMCA_VERMES_43452) and four paratypes (RMCA_VERMES_43453, HU 797, SAMC A094501, MZH https://id.luomus.fi/KV.681)

Symbiotype: MRAC P.70520

Symbioparatypes: MRAC P.70518, MRAC P.98130, MRAC P.98134, MRAC P.7826 

Type host: *Microthrissa congica* (Regan, 1917) (Teleostei, Clupeidae)

Type locality: Mobi River, Democratic Republic of the Congo

Other localities: Poko and Manyanga, Democratic Republic of the Congo

ZooBank registration: The Life Science Identifier (LSID) for *Kapentagyrus bisthoveni* Vanhove, Hermans, Artois & Kmentová n. sp. is urn:lsid:zoobank.org:act:F44D1BAE-04B3-40B4-A306-38AFAF2C74DF.

Material examined: four whole-mounted and one partly mounted specimens.

Infection site: gill filaments

Infection parameters: Poko, Democratic Republic of the Congo: one out of two hosts infected with two worms (infection intensity = 2); Mobi River, Democratic Republic of the Congo: one out of two hosts infected with one worm (infection intensity = 1); Manianga, Democratic Republic of the Congo: two out of eight hosts infected with one worm each (infection intensity = 1); Sankuru River, Democratic Republic of the Congo: single specimen examined was not infected; Atena Island in Kinshasa (then known as Leopoldville), Democratic Republic of the Congo: none out of three hosts infected.

Etymology: the species epithet honours biologist Luc Janssens de Bisthoven (Belgium) for his contributions to both policy development and academic research in the field of scientific capacity building regarding biodiversity in Africa and elsewhere in the Global South.

Diagnosis: a pair of dorsal and ventral anchors each with inner root ca. twice to thrice as long as outer root, short and compact anchor blade. Dorsal and ventral bars V-shaped, enlarged in the middle, similar in branch length and width. Hooks seven pairs. Male copulatory organ with copulatory tube with somewhat triangular basal bulb, gradually narrowing and tapering distally until terminal opening; AP winding once around the copulatory tube from the right side, longer than copulatory tube, reaching its distal end. 

Discussion: like its congener on another representative of *Microthrissa*, *K. chochamandai* n. sp., *K. bisthoveni* n. sp. differs from the Tanganyika species *K. tanganicanus* and *K. limnotrissae* by the gradual transition of the basal bulb towards the rest of the copulatory tube, and from the species infecting members of *Odaxothrissa* in its stubby, compact haptoral anchors. The size of the MCO and the pear-shaped proximal part of the copulatory tube of *K. bisthoveni* n. sp. also remind of those of *K. parisellei* n. sp. infecting *N. parva*. However, contrary to the latter, which displays no heel in the MCO, the basal bulb of *K. bisthoveni* n. sp. has an irregular, heel-like structure, and the inner root of the dorsal anchor of *K. bisthoveni* n. sp. is on average longer than its first hook pair. This aspect also distinguishes *K. bisthoveni* n. sp. from *K. chochamandai* n. sp., and from *K. hugei* n. sp.

#### 3.3.10. Description of *Kapentagyrus boegeri* n. sp.

*Kapentagyrus boegeri* n. sp. (Figure 16; Table 4)

Type material: holotype (RMCA_VERMES_43451) and two paratypes (RMCA_VERMES_43450, UH 814)

Symbiotype: NHMUK 1976.12.20.55

Symbioparatypes: NHMUK 1976.12.20.42-3

Type host: *Microthrissa congica* (Regan, 1917) (Teleostei, Clupeidae)

Type locality: Lualaba River between Kongolo and Kasongo, Democratic Republic of the Congo

ZooBank registration: The Life Science Identifier (LSID) for *Kapentagyrus boegeri* Vanhove, Hermans, Artois & Kmentová n. sp. is urn:lsid:zoobank.org:act:D07DE259-F779-4EF2-9BC1-27273D25EBE8.

Material examined: three whole-mounted specimens.

Infection site: gill filaments

Infection parameters: 12 host specimens investigated, three of which infected by one parasite each (infection intensity = 1).

Etymology: the species epithet honours biologist Walter Boeger (Brazil), professor at the Universidade Federal do Paraná, for his seminal contributions to evolutionary parasitology and infectious diseases research focusing on monogeneans among other non-model organisms.

Diagnosis: a pair of dorsal and ventral anchors each with inner root about double the length of outer root, slender anchor blade. Dorsal and ventral bars V-shaped, similar in branch length and width. Hooks seven pairs. Male copulatory organ with copulatory tube with somewhat elongated basal bulb, tapering distally; AP winding once around the copulatory tube from the left side. 

Discussion: The broad basal bulb of the copulatory tube that gradually transitions into the rest of the copulatory tube sets *K. boegeri* n. sp. aside from species of *Kapentagyrus* infecting hosts belonging to *Odaxothrissa*, *Stolothrissa,* or *Limnothrissa*. The dorsal anchor inner root being on average shorter than the hooks renders *K. boegeri* n. sp. similar to species of *Kapentagyrus* infecting *M. moeruensis* (*K. chochamandai* n. sp.)*, N. parva* (*K. parisellei* n. sp.), and *P. leonensis* (*K. hugei* n. sp., *K. verbisti* n. sp.). In contrast to *K. chochamandai* n. sp., *K. boegeri* n. sp. has a longer point length in the dorsal anchor (7 to 8 µm in *K. boegeri* n. sp. versus 5 to 6 µm in *K. chochamandai* n. sp.). *Kapentagyrus boegeri* n. sp. can be distinguished from *K. parisellei* n. sp. by its longer dorsal anchor at a total length of min. 23 µm, while it is maximally 22 µm long in *K. parisellei* n. sp. From the species infecting *P. leonensis*, *K. boegeri* n. sp. can be distinguished as follows: (1) in contrast to *K. hahni* n. sp., *K. boegeri* n. sp. has a wider incision between the anchor roots; (2) in contrast to *K. hugei* n. sp., with a dorsal bar branch length of max. 22 µm, *K. boegeri* n. sp. has a larger dorsal bar with a branch length of min. 23 µm; (3) in contrast to *K. verbisti* n. sp., the turns in the AP of the MCO are at a sharper angle; and (4) in contrast to *K. pellonulae* with a total length of the dorsal anchor of min. 40 µm and of the ventral anchor of 50 µm, the anchors of *K. boegeri* n. sp. are smaller, at max. 30 µm and 26 µm, respectively.

*Kapentagyrus boegeri* n. sp. differs from *K. bisthoveni* n. sp., which infects the same host species *M. congica* by the more slender anchors, and by the more symmetric appearance of the dorsal anchor roots thanks to a shorter inner root. Therefore, and in contrast to *K. bisthoveni* n. sp., the dorsal anchor inner root is shorter than the hooks.

#### 3.3.11. Description of *Kapentagyrus rochetteae* n. sp.

*Kapentagyrus rochetteae* n. sp. (Figure 17; Table 4)

Type material: holotype (RMCA_VERMES_43454), 10 paratypes (RMCA_VERMES_43455-6, HU 798-9, 815-6, SAMC A094502-3, MZH https://id.luomus.fi/KV.682, https://id.luomus.fi/KV.683)

Symbiotype: MRAC 88001.P.0416

Symbioparatypes: MRAC 88001.P.0417, MRAC 73022.P.0121-3, MRAC 73022.P.0129, MRAC 73022.P.0131

Type host: *Microthrissa royauxi* Boulenger, 1902 (Teleostei, Clupeidae)

Type locality: Pool Malebo, close to Nsele River, Democratic Republic of the Congo

ZooBank registration: The Life Science Identifier (LSID) for *Kapentagyrus rochetteae* Vanhove, Hermans, Artois & Kmentová n. sp. is urn:lsid:zoobank.org:act:65320DF1-6778-42BD-80AB-B425798A7CC2.

Material examined: 11 whole-mounted specimens

Infection site: gill filaments

Infection parameters: eight out of thirteen hosts infected with one or two worms each (infection intensity = 1.4)

Etymology: the species epithet honours bioscience engineer Anne-Julie Rochette (Belgium) for her contributions to capacity development for biodiversity science policy and biodiversity policy in sub-Saharan Africa.

Diagnosis: a pair of dorsal and ventral anchors each with inner root more than twice as long as outer root, short and compact anchor blade. Dorsal and ventral bars V-shaped, enlarged in the middle, with branches of similar length. Hooks seven pairs, short, all similar in length. Male copulatory organ with copulatory tube with basal bulb, quickly narrowing and tapering distally until terminal opening; AP winding once around the copulatory tube, longer than copulatory tube but not always reaching its distal end. 

Discussion: *Kapentagyrus rochetteae* n. sp. resembles *K. bisthoveni* n. sp., *K. chochamandai* n. sp., and *K. parisellei* n. sp. in the compact anchors with a short blade (in contrast to species of *Kapentagyrus* infecting members of *Odaxothrissa* and to *K. boegeri* n. sp. infecting *M. congica*, with their elongated anchors with slender blades). It differs from the former in the shorter hooks of the third, fourth, sixth, and seventh pairs (ca. 7–15 µm in *K. rochetteae* n. sp., versus ca. 15–18 µm in *K. bisthoveni* n. sp.) and the orientation of the loop of the AP (from the left versus from the right in *K. bisthoveni* n. sp.), and from the latter two in the ratio of inner root of dorsal anchor and first pair of hooks (larger than 1 in *K. rochetteae* n. sp. versus below 1 in *K. chochamandai* n. sp. and *K. parisellei* n. sp.).

### 3.4. Identification Key to the Species of Kapentagyrus

Following the description of *Kapentagyrus* by Kmentová et al. [9], which also offers the most recent taxonomic treatment of its species, the following key ascribes much importance to the absolute and proportional lengths of the haptoral hooks and the (roots of the) haptoral anchors. For practical reasons of usability, at times, the host species is also used as a criterion in this key. We refer to the taxonomic account in this study for more details on the diagnostic morphological differences between the species of *Kapentagyrus*.

Length to notch of ventral anchors > 24 µm, and broad inner anchor roots, and total length of dorsal anchors > 31 µm: species infecting members of *Odaxothrissa* → (2)Anchors different → (4)Total length of ventral anchors > 40 µm, inner root length of ventral anchors > 22 µm …… *Kapentagyrus sefcae* n. sp. (parasite of *Odaxothrissa losera*)Total length of ventral anchors < 40 µm, inner root length of ventral anchors < 22 µm → (3)Dorsal anchor with inner root length > 18 µm and outer root length > 7 µm …… *Kapentagyrus marispastoris* n. sp. (parasite of *Odaxothrissa mento*)Dorsal anchor with inner root length < 18 µm and outer root length < 7 µm… … *Kapentagyrus voreli* n. sp. (parasite of *Odaxothrissa ansorgii*) Species infecting *Limnothrissa miodon* or *Stolothrissa tanganicae*, endemic to Lake Tanganyika → (5)Species infecting *Nannothrissa parva*, *Pellonula leonensis*, *Potamothrissa acutirostris,* or members of *Microthrissa*, outside of Lake Tanganyika → (6)Ratio of inner and outer root length of ventral anchors > 2.6, of length to notch of dorsal anchors and the first hook pair < 1.2 and of branch length of dorsal bar and the first pair of hooks < 1.5 …… *Kapentagyrus limnotrissae* (parasite of *Limnothrissa miodon*)Ratio of inner and outer root length of ventral anchors < 2.6, of length to notch of dorsal anchors and the first hook pair > 1.2 and of branch length of dorsal bar and the first pair of hooks > 1.5 … … *Kapentagyrus tanganicanus* (parasite of *Limnothrissa miodon* or *Stolothrissa tanganicae*)Species infecting *Pellonula leonensis* → (7)Species infecting *Nannothrissa parva*, *Potamothrissa acutirostris,* or *Microthrissa* sp. → (10)Total length of dorsal and ventral anchors ≥ 40 µm….… *Kapentagyrus pellonulae* (parasite of *Pellonula leonensis*)Total length of dorsal and ventral anchors < 40 µm → (8)Total length of dorsal anchors ≥ 35 and < 40 µm …… *Kapentagyrus verbisti* n. sp. (parasite of *Pellonula leonensis*)Total length of dorsal anchors < 35 µm → (9) Length to notch of ventral anchors < 19 µm …… *Kapentagyrus hahni* n. sp. (parasite of *Pellonula leonensis*)Length to notch of ventral anchors ≥ 19 µm … … *Kapentagyrus hugei* n. sp. (parasite of *Pellonula leonensis*)Ratio of inner root length of dorsal anchors and first pair of hooks > 1 → (11)Ratio of inner root length of dorsal anchors and first pairs of hooks < 1 → (13)Accessory piece looping copulatory tube from the right side, length of hook pairs HVI and HVII < 15 µm …… *K. rochetteae* n. sp. (parasite of *Microthrissa royauxi*)Accessory piece looping copulatory tube from the left side, length of hook pairs HVI and HVII > 15 µm → (12) Inner root of dorsal anchors < 12 µm …… *Kapentagyrus boegeri* n. sp. (parasite of *Microthrissa congica*)Inner root of dorsal anchors ≥ 12 µm … … *Kapentagyrus bisthoveni* n. sp. (parasite of *Microthrissa congica*)Length to notch of ventral anchor < 19 µm …… *Kapentagyrus parisellei* n. sp. (parasite of *Nannothrissa parva*)Length to notch of ventral anchor > 19 µm …… *Kapentagyrus chochamandai* n. sp. (parasite of *Microthrissa moeruensis* or *Potamothrisssa acutirostris*)

## 4. Discussion

By the examination of 172 individual host clupeids sourced from several biodiversity collections, a total of 112 parasite specimens were retrieved, all belonging to unknown species of *Kapentagyrus*. This allowed the description of 11 species within this genus, more than quadrupling its known species richness from three to fourteen, and increasing the number of pellonuline species from which monogeneans are known from three to eleven. 

### 4.1. Diagnostic Value of Haptoral and Genital Morphology in Kapentagyrus

In distinguishing and delineating species of *Kapentagyrus*, our logic was to focus on features that were confirmed, using sequence data, to be diagnostic in the original description of this genus, including anchor size, anchor root length, hook length, and their ratios. Closely related species of dactylogyrid monogeneans are often distinguished based on their genital hard parts (e.g., [34] for *Cichlidogyrus*). Moreover, other authors have suggested the existence of a phylogenetic signal in the male genital morphology (e.g., [35] for *Characidotrema* Paperna & Thurston, 1968). However, with increased taxon coverage of congeneric dactylogyrid monogeneans in phylogenetic or phylogeographic research, cases have surfaced in which MCO morphology seems conserved across different species. The MCO is so similar between the species of *Kapentagyrus* that it seems of little systematic use: the structural, qualitative/discrete differences are few to none, and quantitative differences in length of the AP or the copulatory tube are not diagnostic [9] (Figure 5). This has also been shown in other dactylogyrid monogeneans. For example, in a molecular phylogenetic analysis, Messu Mandeng et al. [36] found the MCO to hardly change in a lineage of *Cichlidogyrus* even after a distant host-switching event, while barcoding data from Jorissen et al. [37] suggest that *Cichlidogyrus halli* Price & Kirk, 1967 represents a complex in which the species phenotypically differ in the haptor rather than in the MCO. 

In the first-ever taxonomic work in *Kapentagyrus* integrating morphological and molecular data, Kmentová et al. [9] proposed the proportion of the length of the inner and outer roots of the haptoral anchors, and the proportions between the hook length and anchor size, as diagnostic traits to distinguish specimens whose identity had been confirmed by sequence data. Therefore, in describing 11 new species here, we focused on these same features used by Kmentová et al. [9]. This aligns with the importance of the length of the outer roots in the anchors, and of hook lengths, to separate parasites in the PCA biplot (Figure 3b and Figure 4b). On the other hand, the multivariate statistical analyses also show that species of *Kapentagyrus* cannot be completely separated on the basis of morphometrics alone. Comparing the variability in the haptoral characters of the species described here, we propose that other haptoral features, such as the angle between the inner and outer roots of the anchors, the width of the anchor roots, and the angle between the anchor basis and its blade, will be interesting to consistently consider in taxonomic work within *Kapentagyrus*. We also suggest that the maximal width of the basal bulb of the copulatory tube, and the maximal width of the rest of the copulatory tube, should be considered.

### 4.2. Distribution and Species Richness of Members of Kapentagyrus on Pellonuline Hosts

In several cases, species-level distinctions based on haptoral characters seem to go together with geographical separation. While *K. voreli* n. sp. and *K. marispastoris* n. sp. infect different hosts in resp. Gabon (*O. ansorgii*) and Ghana (*O. mento*), *K. bisthoveni* n. sp. and *K. boegeri* n. sp. both occur on *M. congica*, with the latter being described from the more upper reaches of the Congo Basin. On the other hand, the situation in the Volta Basin, where *P. leonensis* is infected by *K. hugei* n. sp., *K. hahni* n. sp. and *K. pellonulae*, suggests that, just as in Lake Tanganyika, conspecific hosts may harbour several species of *Kapentagyrus* even in the same area. Given our opportunistic collection-based sampling, these three species were so far not found in exactly the same locality at the same time. As there are no clear physical geographic barriers, perhaps differences in physicochemical environmental conditions in different parts of the Volta Basin influence parasite species distribution. Hence, it will be interesting to investigate the spatial and temporal patterns in their communities. Indeed, Kmentová [38] showed that different species of *Kapentagyrus* co-infecting a host are not randomly distributed. In the work of Kmentová [38], spatial or temporal separation between parasite species on the same host species *L. miodon* may explain the sympatric occurrence of *K. limnotrissae* and *K. tanganicanus*, species with morphologically near-identical genitals. The reported absence in the present study of monogeneans on the gills of *L. miodon* from man-made Lake Itezhi-Tezhi, where this fish was introduced in 1992 to fill the vacant pelagic niche, can probably be explained when assuming fry was used for this translocation (following the reasoning in [15]). We reiterate that parasitological data can be informative in the context of data-poor fisheries systems based on non-native species. Therefore, intensified parasitological screening is recommended for Lake Itezhi-Tezhi, from which hardly any published information is available regarding the introduction and establishment of *L. miodon* [39]. 

A definitive species inventory is obviously impossible because of the low sample sizes frequently encountered in opportunistic collection-based sampling. It is noteworthy that, so far, only the Tanganyika endemic *L. miodon* [9], *P. leonensis* from West Africa and *M. congica* from the Congo Basin harbour more than a single species of *Kapentagyrus*. For *L. miodon,* this was attributed by Kmentová et al. [26] to a recent host-switch, probably linked to the sympatric occurrence and predator–prey relationship of the two pellonuline sister species in Lake Tanganyika. To explain the high species richness of *Kapentagyrus* on *P. leonensis*, known to harbour four nominal species so far, we refer to Pariselle et al. [40], whose comparative study of the monogenean gill parasite fauna of West African “tilapias” led them to suggest a large distribution range of the host and the presence of related host species as factors increasing parasite species richness in a given host species. Indeed, *P. leonensis* is widely distributed throughout West and West-Central Africa. It also co-occurs with other freshwater clupeids, for instance, in Lake Volta, where three clupeids exist [41]. A similar reasoning may apply to *M. congica*, which, in our limited sampling, seems to host *K. bisthoveni* n. sp. and *K. boegeri* n. sp., respectively, in different parts of the Congo Basin.

### 4.3. Research Perspectives Regarding Taxonomy and Speciation within Kapentagyrus

The mechanisms behind host sharing and host-specificity in monogeneans are poorly understood and general patterns are hard to deduce from our currently highly fragmented knowledge. Arguably, the best-studied African freshwater fish parasites in this regard belong to another dactylogyrid genus, *Cichlidogyrus*. In the latter lineage, Cruz-Laufer et al. [42] propose that both ecological opportunity and the phylogenetic history of the hosts determine the host range. The role of ecological opportunity is aptly illustrated here by the fact that host-switching does not necessarily lead to speciation in *Kapentagyrus*. Indeed, in Lake Tanganyika, *L. miodon* and *S. tanganicae* share *K. tanganicanus*, and sympatric *M. moeruensis* and *P. acutirostris* in Lake Mweru are infected by the same species of *Kapentagyrus*, *K. chochamandai* n. sp. The role of host interrelationships may be illustrated by the close morphological similarity of the monogeneans infecting species of *Odaxothrissa*: their dorsal anchors are highly divergent when compared to the other species described here (Figure 3a and Figure 4a). However, this is a preliminary conclusion; African pellonuline clupeids are probably in dire need of a taxonomic revision as not all genera are monophyletic [6,43]. Whatever the case, we support the view of Pariselle et al. [40] that a comparative study of the species richness and distribution patterns in comparable and species-rich host-parasite systems is the way forward to discern the general patterns of parasite speciation. We hope that the present study can garner attention for *Kapentagyrus*. This recently described genus would, in our view, provide interesting insights when compared to *Cichlidogyrus*, the lineage of mainly cichlid-infecting monogeneans proposed as a model for parasite macro-evolution by Cruz-Laufer et al. [44]. For example, contrasting these genera would provide interesting insights in the evolution and diagnostic value of the attachment and reproductive organs in parasites (see examples above).

Understanding the evolutionary history of the gill parasite fauna of pellonuline clupeids would be highly informative on the biogeographical patterns of marine–freshwater transitions into continental Africa, and the influence these have on parasite speciation, as suggested for cichlids by Pariselle et al. [17]. In contexts where coastal waters are shaped by a complex hydrogeology, genetic data on monogenean parasites may even provide a higher resolution than fish genetics when dealing with populations inhabiting basins that drain into adjacent estuaries. This was demonstrated by Vanhove et al. [45] for the gyrodactylid *Gyrodactylus benedeni* Vanhove & Huyse, 2014 infecting Western Greece goby *Economidichthys pygmaeus* (Holly, 1929). A better understanding of the phylogeny of *Kapentagyrus* may, therefore, shed light on the sequence of events leading to the colonisation of African inland waters by pellonulines, and on standing questions, e.g., their putative sister-group relationship with the catadromous Bonga shad *Ethmalosa fimbriata* (Bowdich, 1825) (see [43]). As these research questions require a robust phylogenetic framework for the order and timing of diversification events, a major obstacle for developing these sardine parasites as a model is the near absence of genetic data. Sequence information exists only for the Lake Tanganyika endemics within *Kapentagyrus* [9,26]. It is indeed a limitation of our current approach that it has hitherto proven impossible to obtain sequence data from dactylogyrid monogenean specimens stemming from most historical fish collections (e.g., [46]) in view of the commonly used original fixation in formaldehyde. This, however, need not hamper species identification and description, as demonstrated by Jorissen et al. [20], for species of *Cichlidogyrus*, and by Kmentová et al. [15] for members of *Kapentagyrus*. Nevertheless, voucher specimens of the retrieved hologenophores of the presently newly described species of *Kapentagyrus* have been preserved for the possible recovery of short fragments of systematically informative markers in the future [47,48].

## 5. Conclusions

Based on haptoral features that were proven to be diagnostic at the time of the original description of *Kapentagyrus*, we describe 11 new species of *Kapentagyrus* from nine pellonuline clupeid host species. Especially, the size of the haptoral structures, in particular the (proportional) lengths of the anchor roots and hooks, appear useful to distinguish species within this dactylogyrid genus. While many pellonulines seem to be infected by a single species of *Kapentagyrus* on their gills, two host species that we sampled over a geographically large range, *M. congica* and *P. leonensis*, harbour several species. Conversely, sympatric *M. moeruensis* and *P. acutirostris* share a species of *Kapentagyrus*. Additional sampling efforts may shed light on whether this indeed indicates a pattern of increased parasite species richness in allopatry, and host-switching in sympatry. Meanwhile, we hope that the present study will be helpful in identifying these morphologically highly similar monogeneans, will underscore the potential of collection-based helminthological research, and will stimulate further research on these understudied parasites of economically and ecologically important African fishes.

## Figures and Tables

**Figure 1 animals-11-03578-f001:**
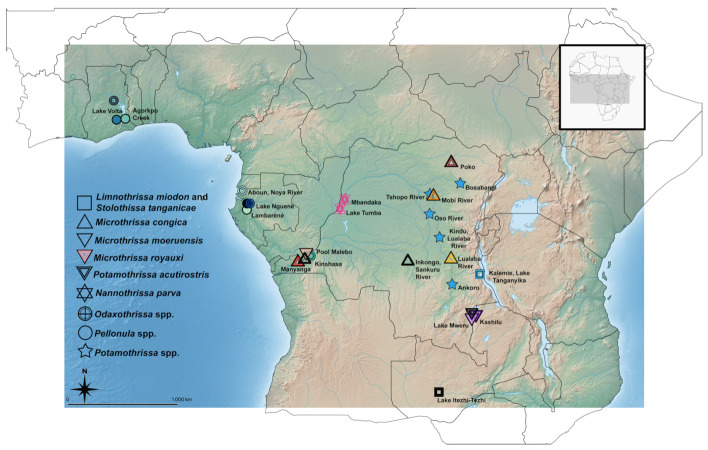
Overview of the sampling localities of freshwater African clupeid species examined for the presence of monogenean parasites as part of this study. Colours and signs denote parasite species, host species, and locality of origin displayed in the subsequent results of statistical analyses.

**Figure 2 animals-11-03578-f002:**
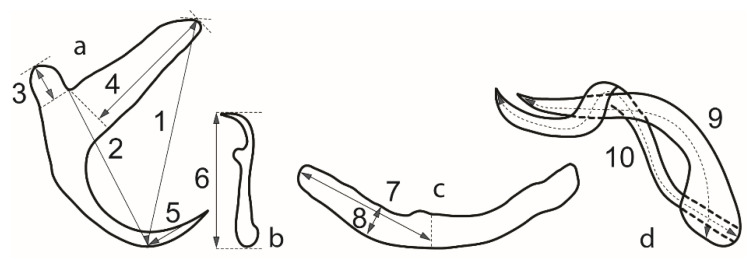
Measurements for hard parts of haptor and genitals of species of *Kapentagyrus* spp. (a) Anchor, 1-total length. 2-length to notch. 3-outer root length. 4-inner root length. 5-point length. (b) Hook, 6-length. (c) Bar, 7-branch length. 8-branch width. (d) Male copulatory organ, 9-copulatory tube axial length. 10-accessory piece axial length.

**Figure 3 animals-11-03578-f003:**
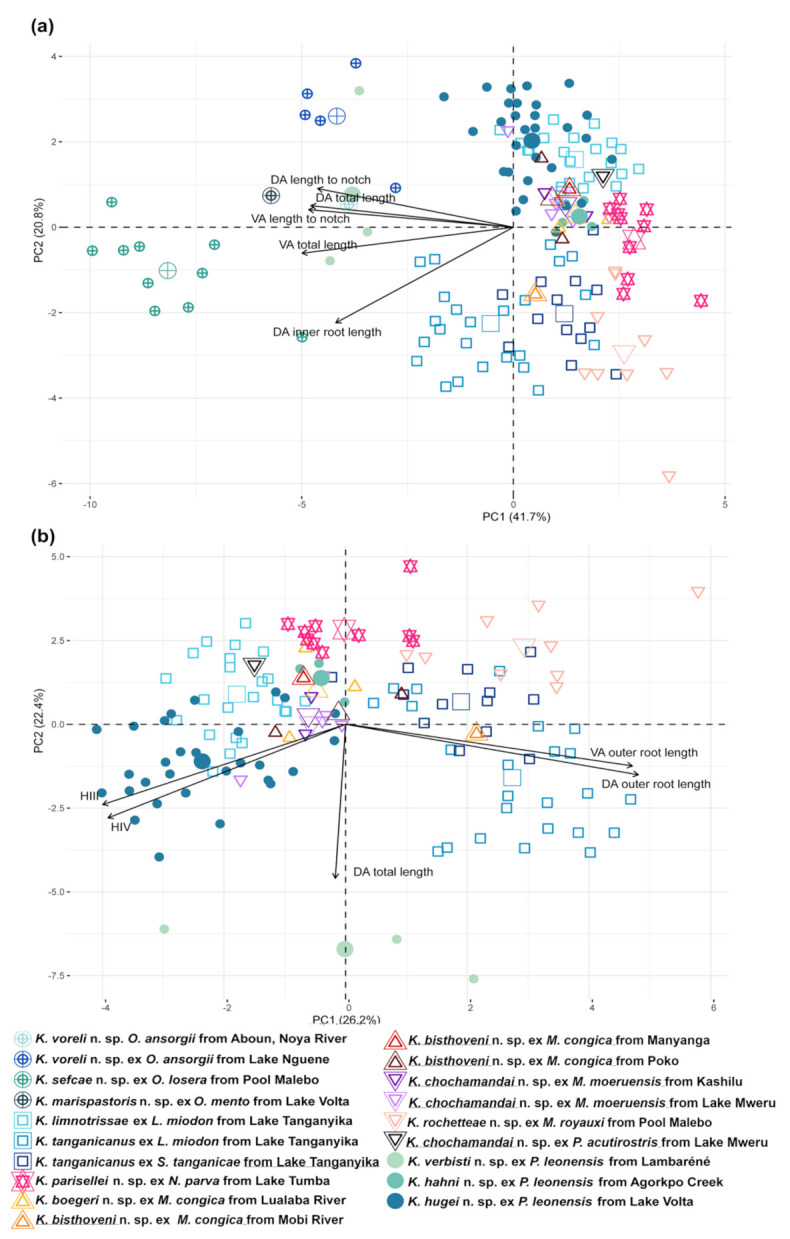
Morphometric variability of haptoral structures of *Kapentagyrus* spp. (**a**) Biplot of PCA (first two axes) including all species of *Kapentagyrus*. (**b**) Biplot of PCA (first two axes) excluding species infecting members of *Odaxothrissa*. Colours and signs denote parasite species, host species, and locality of origin.

**Figure 4 animals-11-03578-f004:**
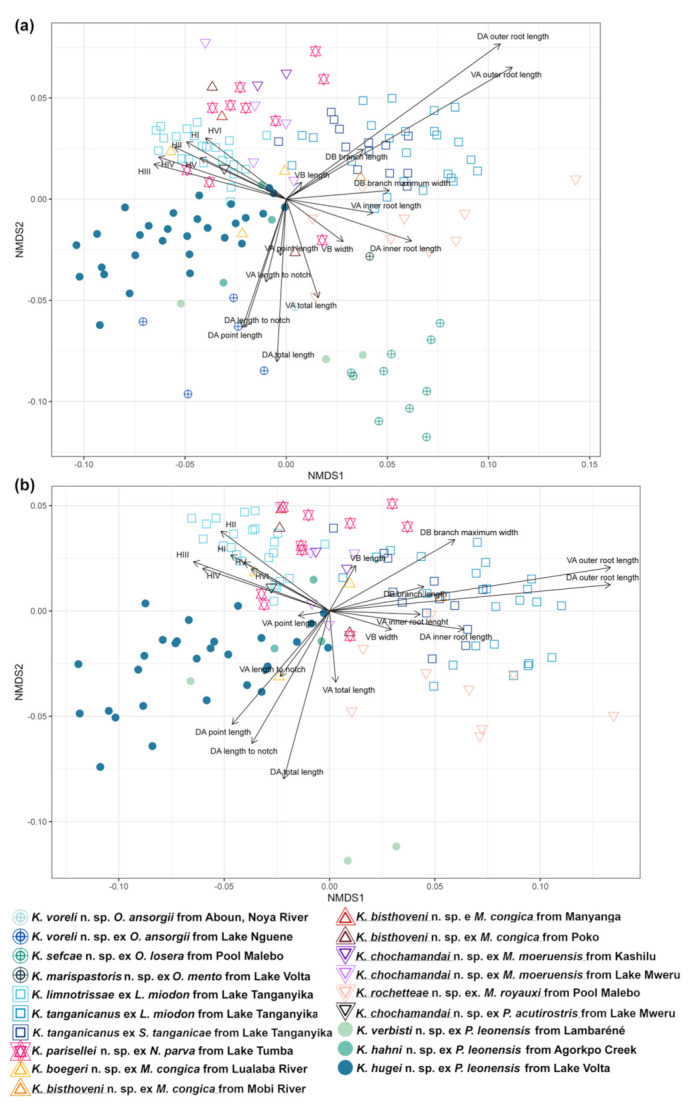
Morphometric variability of haptoral structures of *Kapentagyrus* spp. (**a**) Biplot of NMDS including all species of *Kapentagyrus*. (**b**) Biplot of NMDS excluding species infecting members of *Odaxothrissa*. Colours and signs denote parasite species, host species, and locality of origin.

**Figure 5 animals-11-03578-f005:**
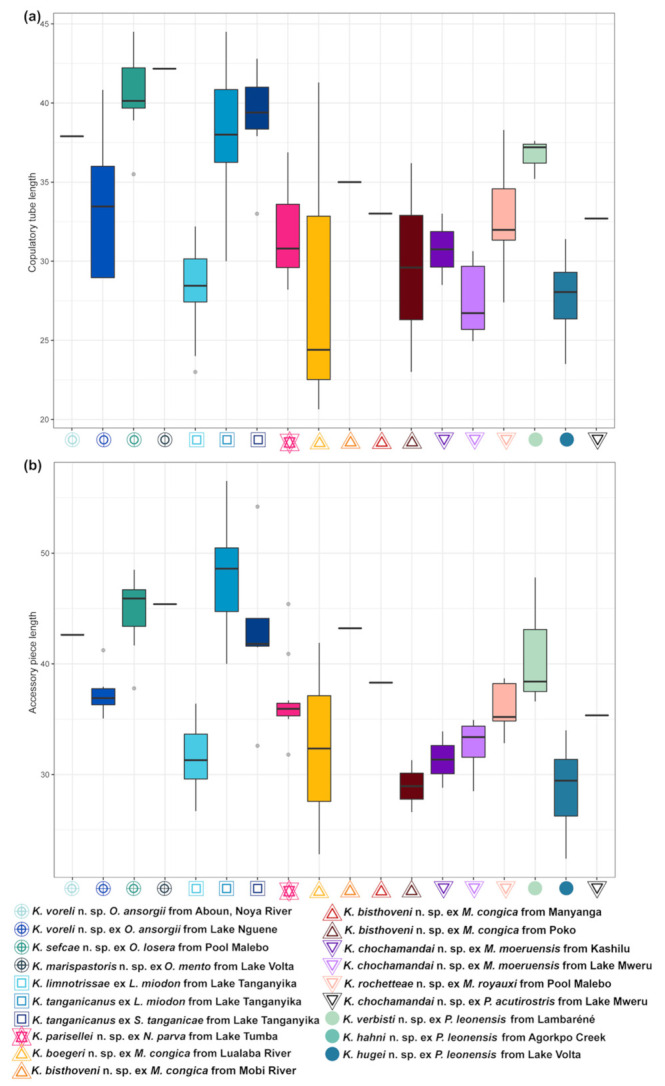
Morphometric variability of copulatory organ structures of *Kapentagyrus* spp. (**a**) Box-plot graph with copulatory tube length (y-axis in μm). (**b**) Box-plot graph with accessory piece length (y-axis in μm). Colours and signs denote parasite species, host species, and locality of origin.

**Figure 6 animals-11-03578-f006:**
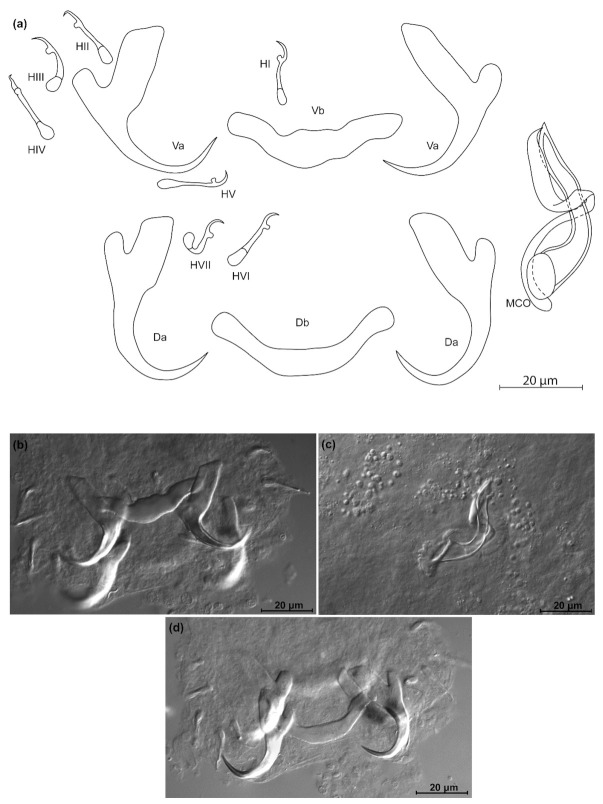
*Kapentagyrus voreli* n. sp. (ex *Odaxothrissa ansorgii*). (**a**) Drawings of sclerotized structures, Va-ventral anchors. Da-dorsal anchors. Db-dorsal bar. Vb-ventral bar. H-hooks (pairs I to V—ventral; pairs VI, VII—dorsal). MCO-male copulatory organ. (**b**) Micrograph of ventral anchors and ventral bar. (**c**) Micrograph of MCO. (**d**) Micrograph of dorsal anchors and dorsal bar.

**Figure 7 animals-11-03578-f007:**
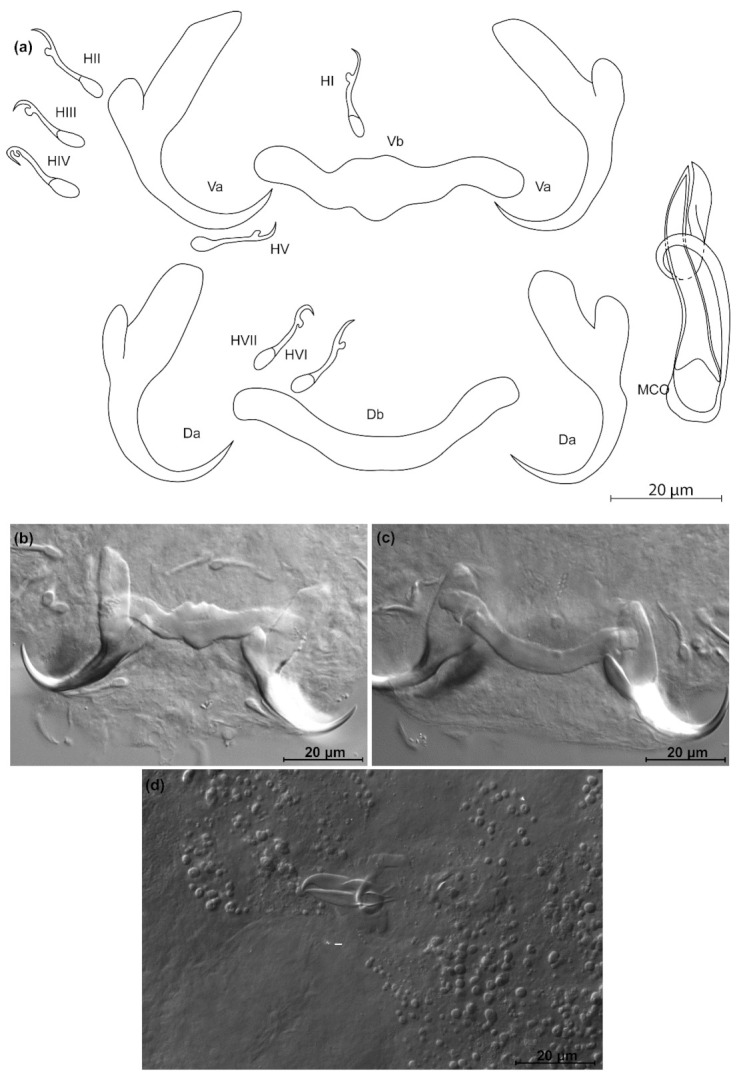
*Kapentagyrus marispastoris* n. sp. (ex *Odaxothrissa mento*). (**a**) Drawings of sclerotized structures, Va-ventral anchors, Da-dorsal anchors, Db-dorsal bar, Vb-ventral bar, H-hooks (pairs I to V—ventral; pairs VI, VII—dorsal), MCO-male copulatory organ. (**b**) Micrograph of ventral anchors and ventral bar. (**c**) Micrograph of dorsal anchors and dorsal bar. (**d**) Micrograph of MCO.

**Figure 8 animals-11-03578-f008:**
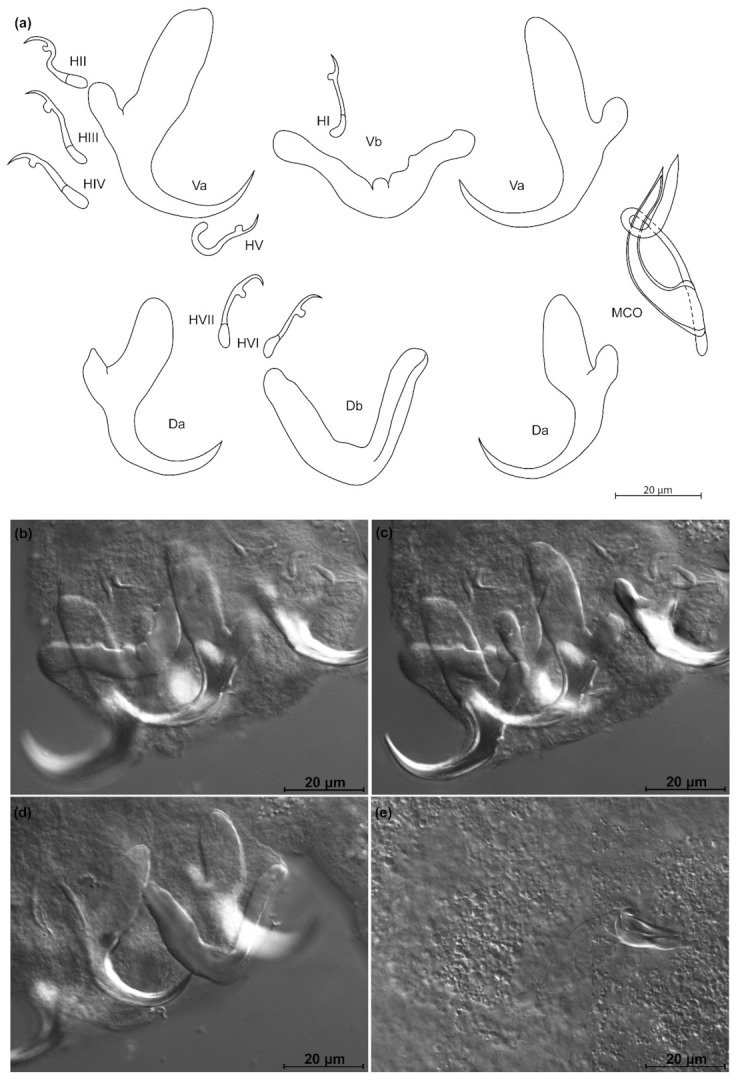
*Kapentagyrus sefcae* n. sp. (ex *Odaxothrissa losera*). (**a**) Drawings of sclerotized structures, Va-ventral anchors, Da-dorsal anchors, Db-dorsal bar, Vb-ventral bar, H-hooks (pairs I to V—ventral; pairs VI, VII—dorsal), MCO-male copulatory organ. (**b**) Micrograph of ventral bar. (**c**) Micrograph of ventral anchors. (**d**) Micrograph of dorsal anchors and dorsal bar. (**e**) Micrograph of MCO.

**Figure 9 animals-11-03578-f009:**
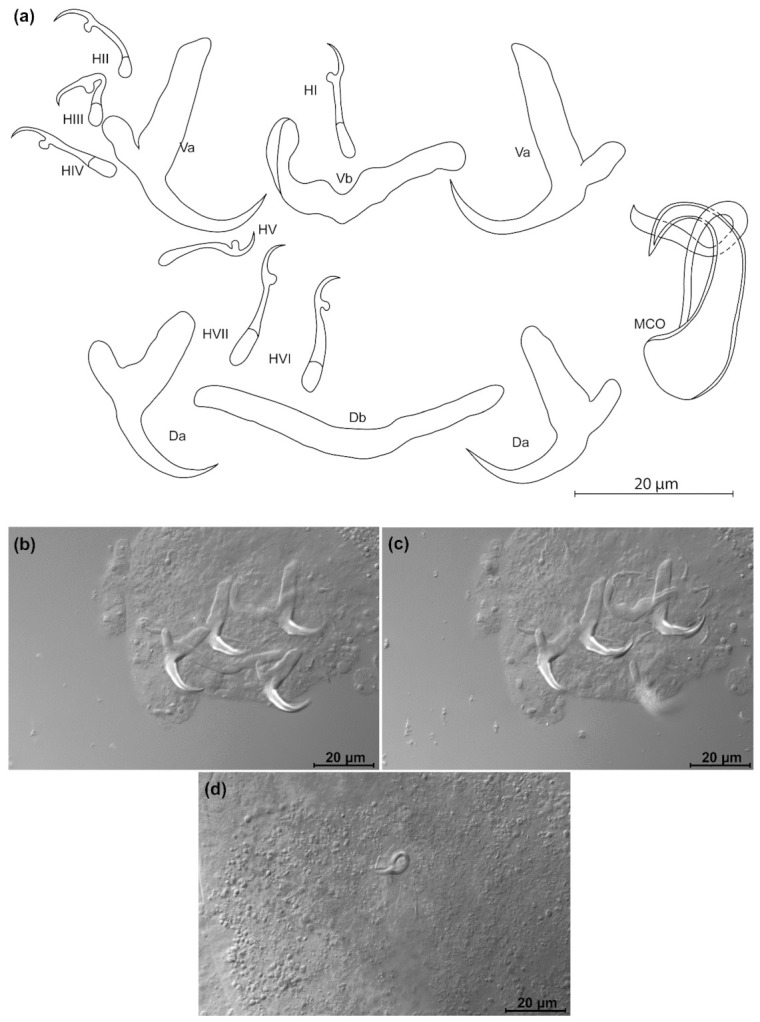
*Kapentagyrus parisellei* n. sp. (ex *Nannothrissa parva*). (**a**) Drawings of sclerotized structures, Va-ventral anchors, Da-dorsal anchors, Db-dorsal bar, Vb-ventral bar, H-hooks (pairs I to V—ventral; pairs VI, VII—dorsal), MCO-male copulatory organ. (**b**) Micrograph of dorsal and ventral anchors, dorsal bar. (**c**) Micrograph of ventral anchors and ventral bar. (**d**) Micrograph of MCO.

**Figure 10 animals-11-03578-f010:**
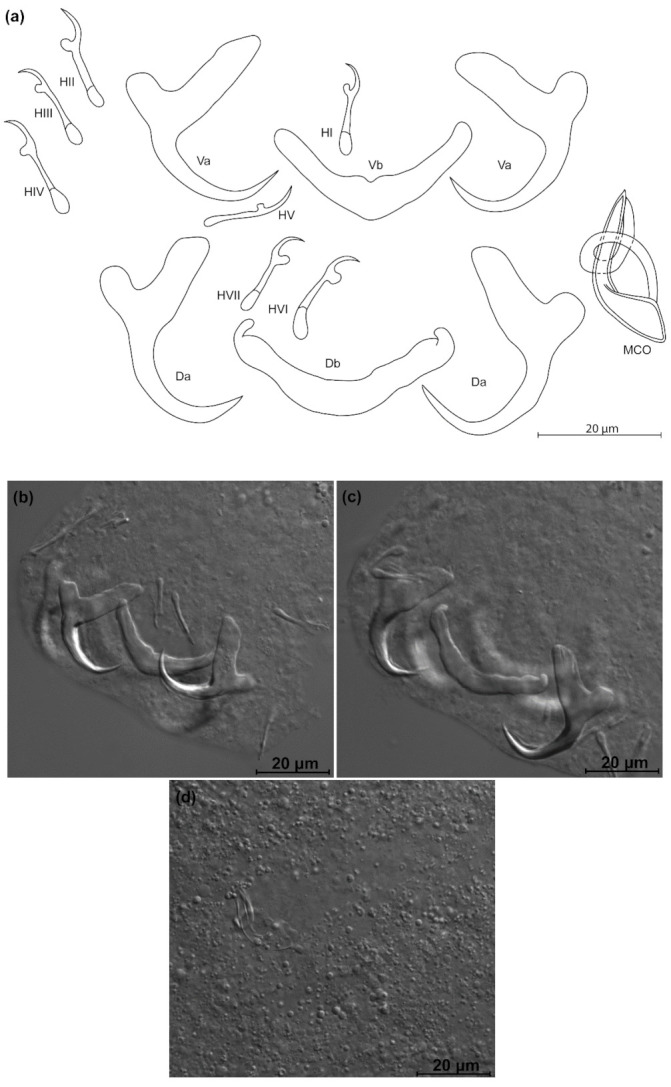
*Kapentagyrus hugei* n. sp. (ex *Pellonula leonensis*). (**a**) Drawings of sclerotized structures, Va-ventral anchors, Da-dorsal anchors, Db-dorsal bar, Vb-ventral bar, H-hooks (pairs I to V—ventral; pairs VI, VII—dorsal), MCO-male copulatory organ. (**b**) Micrograph of ventral anchors and ventral bar. (**c**) Micrograph of dorsal anchors and dorsal bar. (**d**) Micrograph of MCO.

**Figure 11 animals-11-03578-f011:**
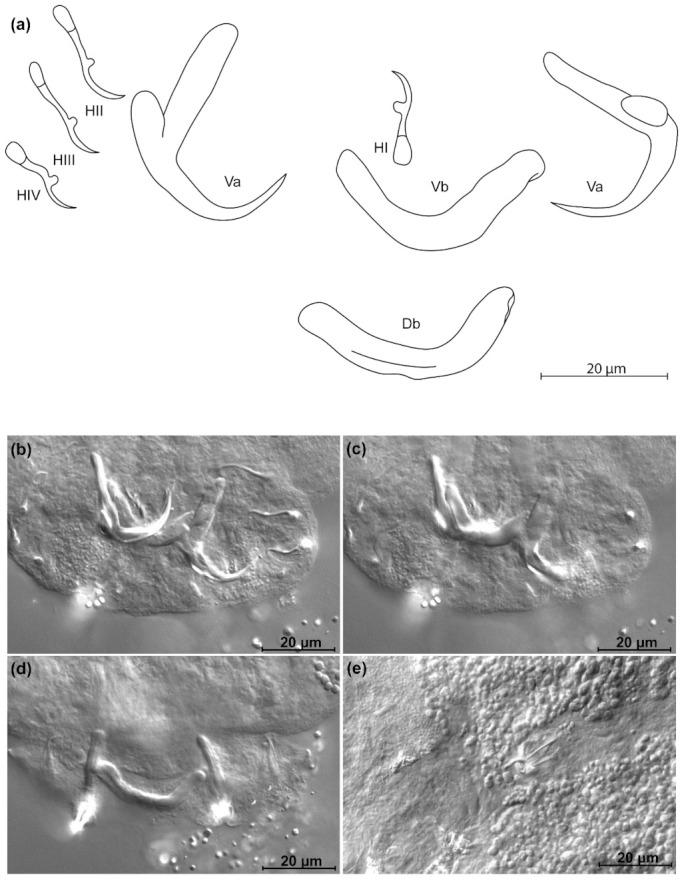
*Kapentagyrus hahni* n. sp. (ex *Pellonula leonensis*). (**a**) Drawings of sclerotized structures, Va-ventral anchors, Db-dorsal bar, Vb-ventral bar, H-hooks (pairs I to IV—ventral). (**b**) Micrograph of ventral anchors and ventral bar. (**c**) Micrograph of dorsal anchors and dorsal bar. (**d**) Micrograph of dorsal anchors and dorsal bar. (**e**) Micrograph of MCO.

**Figure 12 animals-11-03578-f012:**
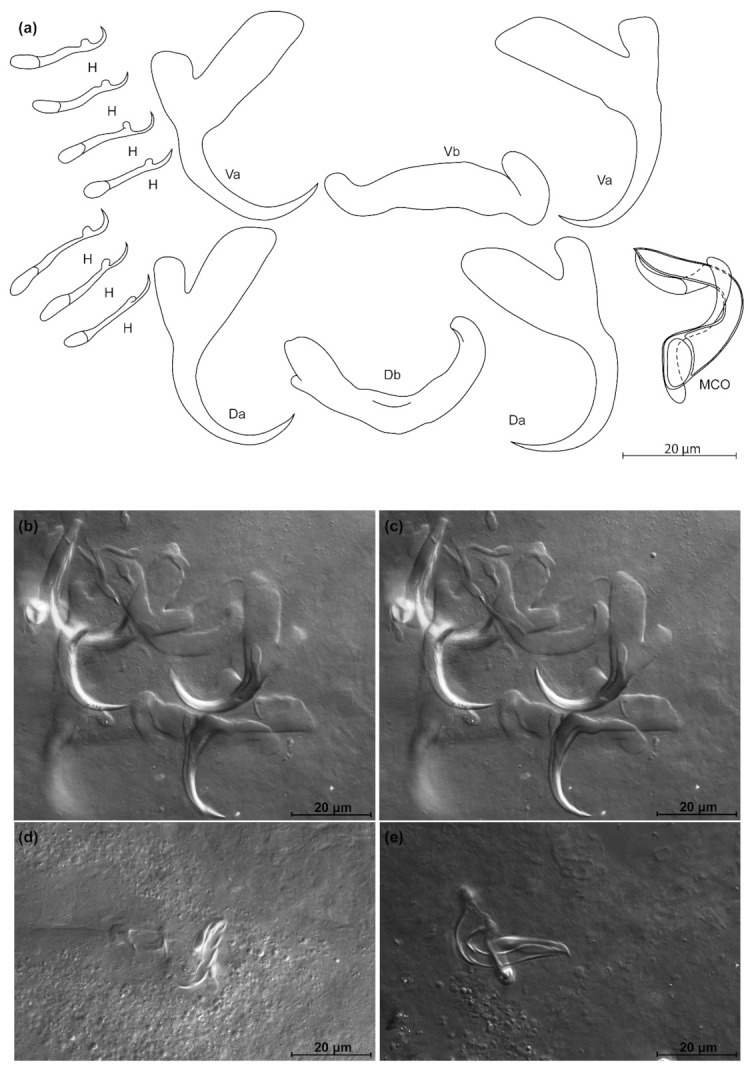
*Kapentagyrus verbisti* n. sp. (ex *Pellonula leonensis*). (**a**) Drawings of sclerotized structures, Va-ventral anchors, Da-dorsal anchors, Db-dorsal bar, Vb-ventral bar, H-hooks (hook pair not identified), MCO-male copulatory organ. (**b**) Micrograph of dorsal anchors and dorsal bar. (**c**) Micrograph of ventral anchors and ventral bar. (**d**) Micrograph of MCO. (**e**) Micrograph of MCO (not from the holotype).

**Figure 13 animals-11-03578-f013:**
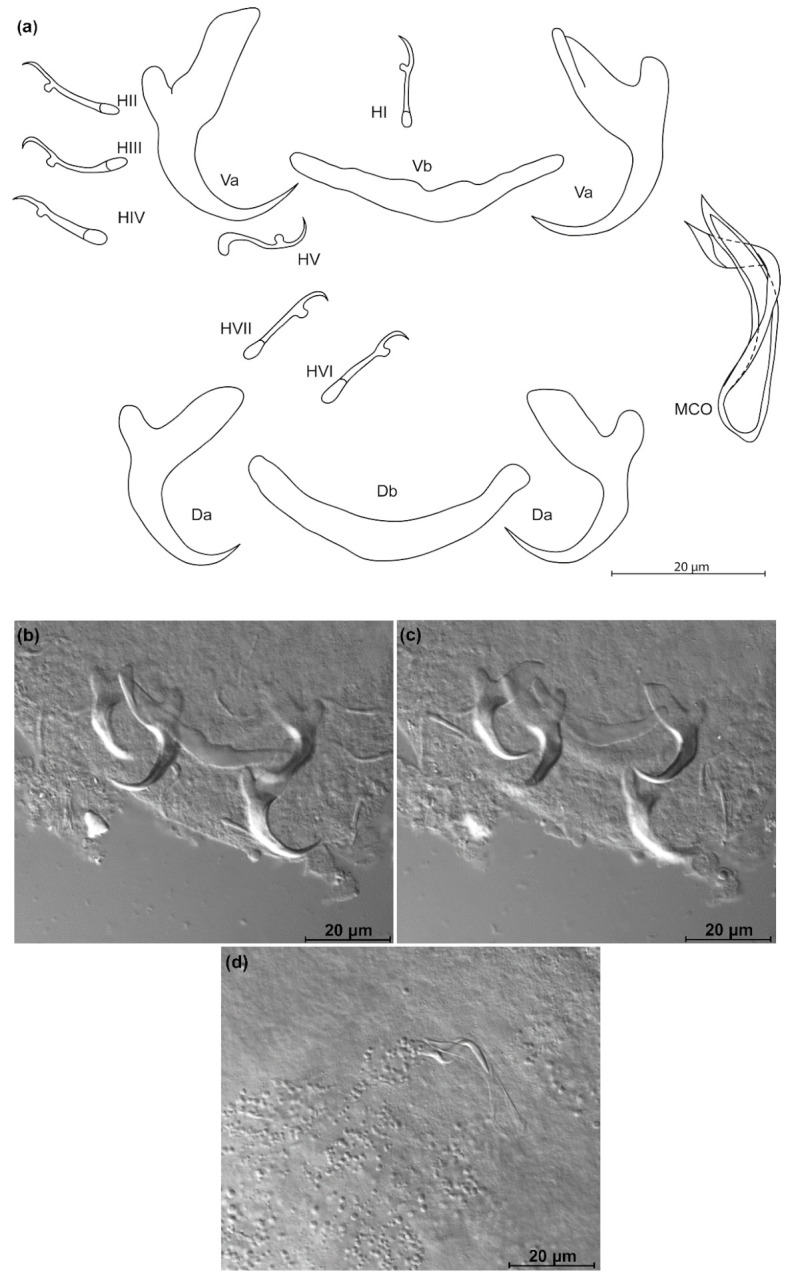
*Kapentagyrus chochamandai* n. sp. (ex *Microthrissa moeruensis*). (**a**) Drawings of sclerotized structures, Va-ventral anchors, Db-dorsal bar, Vb-ventral bar, H-hooks (pairs I to V—ventral; pairs VI, VII—dorsal). (**b**) Micrograph of ventral anchors and ventral bar. (**c**) Micrograph of dorsal anchors and dorsal bar. (**d**) Micrograph of MCO.

**Figure 14 animals-11-03578-f014:**
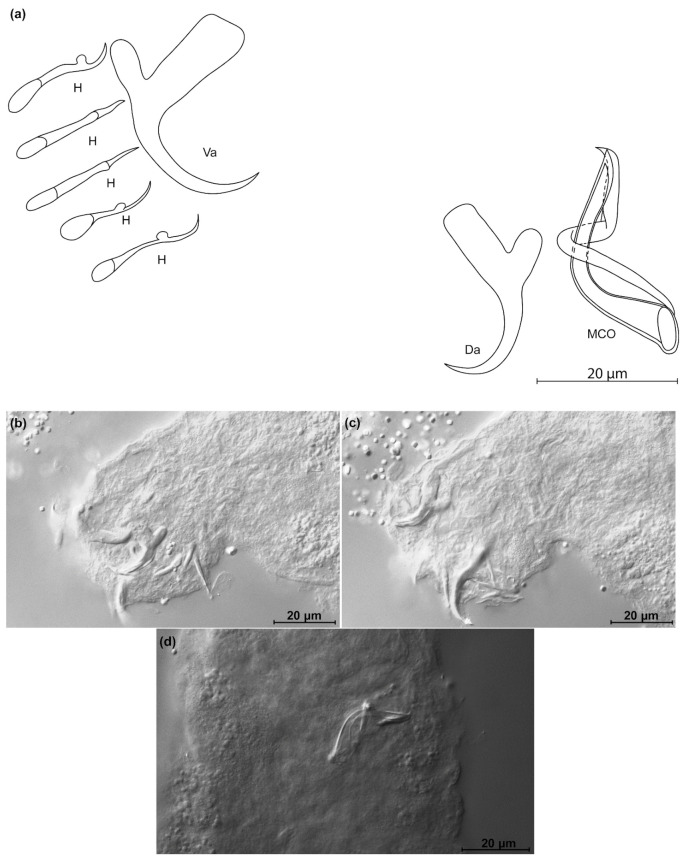
*Kapentagyrus chochamandai* n. sp. (ex *Potamothrissa acutirostris*). (**a**) Drawings of sclerotized structures, Va-ventral anchors, Db-dorsal bar, Vb-ventral bar, H-hooks (hook pairs not identified). (**b**) Micrograph of ventral anchor and ventral bar. (**c**) Micrograph of dorsal anchor. (**d**) Micrograph of MCO.

**Figure 15 animals-11-03578-f015:**
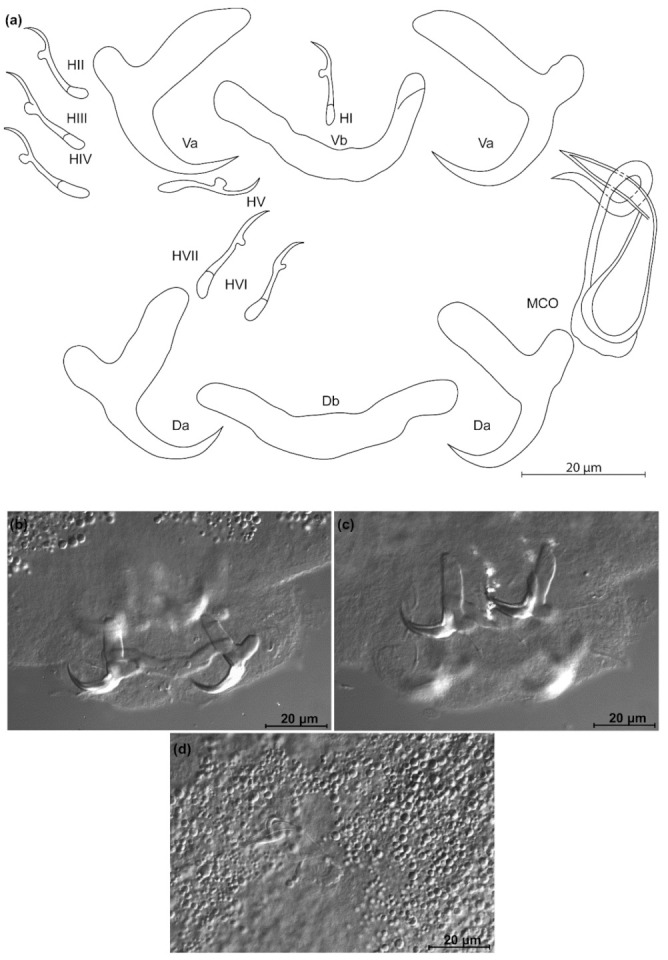
*Kapentagyrus bisthoveni* n. sp. (ex *Microthrissa congica*). (**a**) Drawings of sclerotized structures, Va-ventral anchors, Db-dorsal bar, Vb-ventral bar, H-hooks (pairs I to V—ventral; pairs VI, VII—dorsal). (**b**) Micrograph of dorsal anchors and dorsal bar. (**c**) Micrograph of ventral anchors and ventral bar. (**d**) Micrograph of MCO.

**Figure 16 animals-11-03578-f016:**
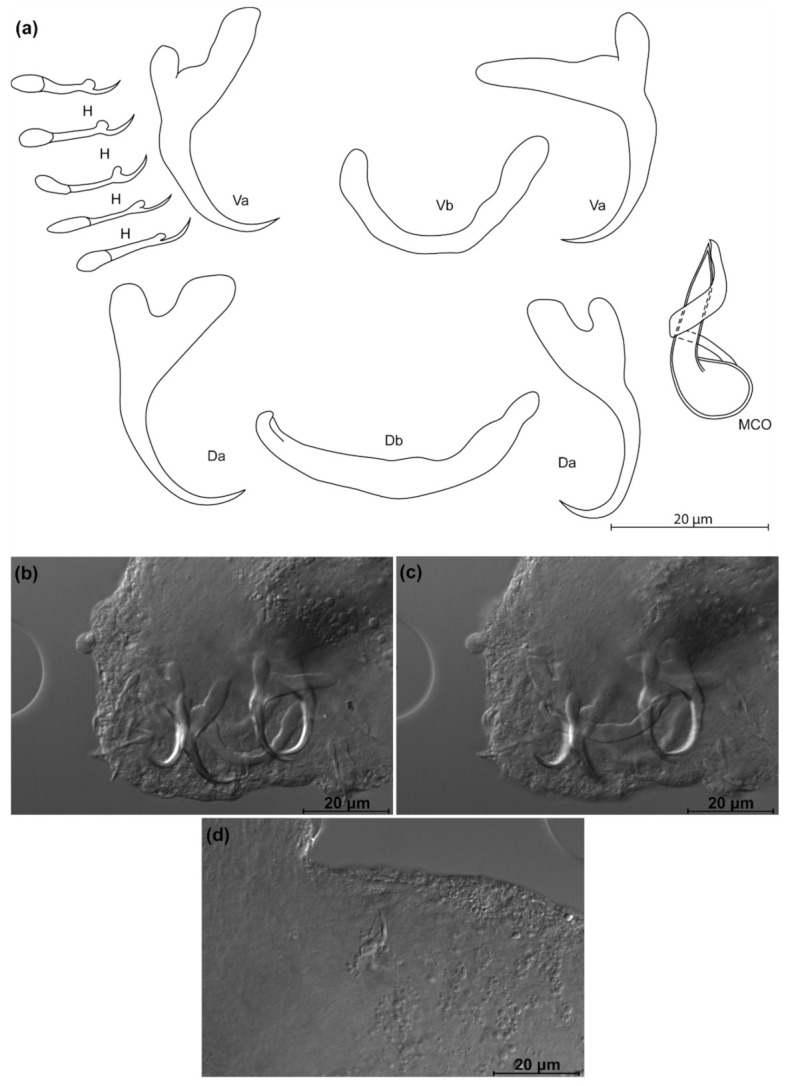
*Kapentagyrus boegeri* n. sp. (ex *Microthrissa congica*). (**a**) Drawings of sclerotized structures, Va-ventral anchors, Db-dorsal bar, Vb-ventral bar, H-hooks (no pairs assigned). (**b**) Micrograph of ventral anchors and ventral bar. (**c**) Micrograph of dorsal anchors and dorsal bar. (**d**) Micrograph of MCO.

**Figure 17 animals-11-03578-f017:**
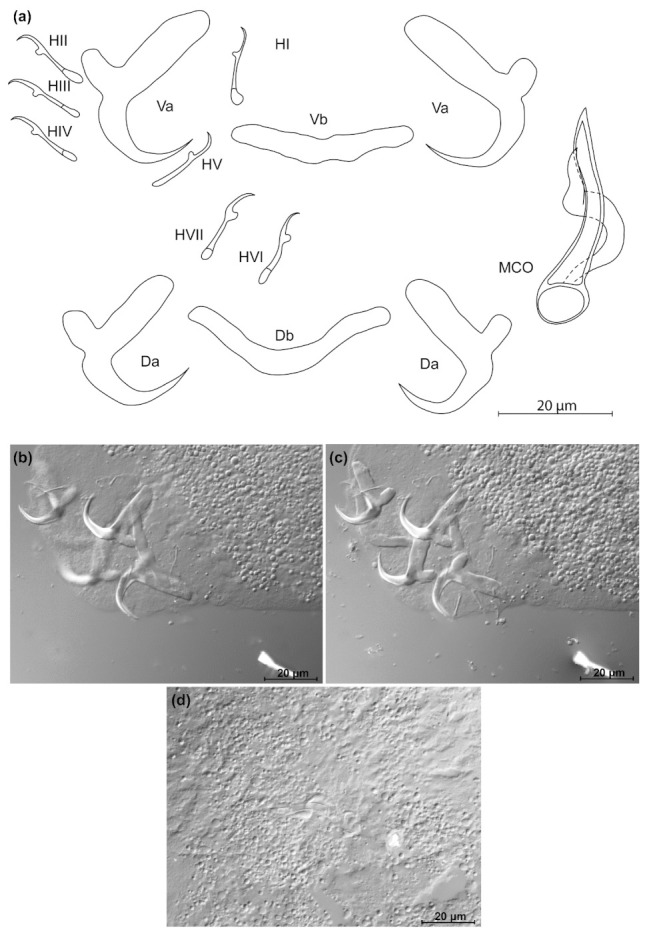
*Kapentagyrus rochetteae* n. sp. (ex *Microthrissa royauxi*). (**a**) Drawings of sclerotized structures, Va-ventral anchors, Db-dorsal bar, Vb-ventral bar, H-hooks (pairs I to V—ventral; pairs VI, VII—dorsal). (**b**) Micrograph of ventral anchors and ventral bar. (**c**) Micrograph of dorsal anchors and dorsal bar. (**d**) Micrograph of MCO.

**Table 1 animals-11-03578-t001:** Sampling details of the pellonuline host specimens investigated.

Host Species	Museum Accession Number	Number of Specimens Examined	Locality	Country	Latitude	Longitude	Date
*Limnothrissa miodon*		24	Lake Itezhi-Tezhi	Zambia	−15.753514	25.969534	17 May 2018
*Microthrissa congica*	MRAC P.98042-98135	9	Manyanga	DRC	−4.9	14.38333	24 September 1954
*Microthrissa congica*	MRAC P.7806-7828	2	Poko	DRC	3.15	26.88333	1912
*Microthrissa congica*	MRAC P.70510-70520	2	Mobi River	DRC	0.4	25.43333	31 July 1947
*Microthrissa congica*	MRAC P.51337-51345	1	Inkongo, Sankuru River	DRC	−4.88333	23.26667	1937
*Microthrissa congica*	MRAC P.120555-120576	3	Kinshasa	DRC	−4.31667	15.31667	12 October 1948
*Microthrissa congica*	NHMUK 1976.12.20.42-77	12	Lualaba River	DRC	−4.781850	26.884137	unknown
*Microthrissa moeruensis*	MRAC 1993.145.P.0033-0064	11	Lake Mweru	Zambia	−9.43333	28.71667	1993
*Microthrissa moeruensis*	MRAC 1994.019.P.2022-2080	5	Kashilu	Zambia	−9.43	28.72	4 August 1993
*Microthrissa royauxi*	MRAC 73022.P.0037-0131	11	Pool Malebo	DRC	−4.1	15.25	23 September 1957
*Microthrissa royauxi*	MRAC 88001.P.0407-0417	2	Pool Malebo	DRC	−4.33	15.38333	13 September 1957
*Nannothrissa parva*	MRAC P.93560-93620	9	Lake Tumba	DRC	−0.61667	17.81667	December 1953
*Nannothrissa parva*	MRAC P.100646-100655	10	Lake Tumba	DRC	−0.61667	17.81667	29–30 September 1955
*Nannothrissa parva*	MRAC P.430-440	2	Mbandaka	DRC	0.06667	18.26667	15 May 1905
*Odaxothrissa mento*	MRAC 1973.007.P.0019-0026	2	Lake Volta	Ghana	8.21667	−0.65	22 May 1968
*Odaxothrissa mento*	MRAC 93-127-P-0003-0009	1	Lake Volta	Ghana	8.21667	−0.65	19 May 1993
*Odaxothrissa ansorgi*	MRAC A0-048-P-1217-1261	10	Lake Nguene	Gabon	−0.18983	10.47233	30 August–1 September 1999
*Odaxothrissa ansorgi*	MRAC A1-070-P-0266	2	Aboun, Noya River	Gabon	0.86667	9.85	9 February 2001
*Odaxothrissa losera*	MRAC 1977.042.P.0001-0011	2	Pool Malebo	DRC	−4.1	15.25	April 1977
*Potamothrissa acutirostris*	MRAC P.124782-124799	2	Ankoro	DRC	−6.75	26.95	18 March 1947
*Potamothrissa acutirostris*	NHMUK 1920.5.26.3-12	10	Kilwa, Lake Mweru	DRC	−9.28357	28.32238	unknown
*Potamothrissa acutirostris*	MRAC 1989.043.P.0283-0290	2	Tshopo River	DRC	0.55	25.11667	31 March 1989
*Potamothrissa acutirostris*	MRAC P.8115-8120	1	Bosabangi	DRC	1.45	27.61667	1912
*Potamothrissa obtusirostris*	MRAC P.70319-70326	1	Kindu, Lualaba River	DRC	−2.95	25.93333	21 July 1947
*Potamothrissa obtusirostris*	MRAC 1989.043.P.0271-0276	1	Oso River	DRC	−1.05	25.11667	18 December 1988
*Pellonula leonensis*	NHMUK 1964.10.12.23-29	7	Agorkpo Creek	Ghana	6.715130	0.309700	unknown
*Pellonula leonensis*	MRAC 1973.005.P.0285-0373	19	Lake Volta	Ghana	6.66667	−0.41667	30 September 1966
*Pellonula leonensis*	MRAC 2000.048.P.1084-1123	6	Lambaréné	Ghana	−0.7	10.21667	17 January 2000
*Pellonula vorax*	MRAC P.71028-71040	2	Lake Nguene	Gabon	−0.18983	10.4723	30 August–1 September 1999
*Pellonula vorax*	MRAC 2000.048.P.1155-1162	1	Lake Nguene	Gabon	−0.18983	10.4723	August 1947

**Table 2 animals-11-03578-t002:** Morphometric data (µm) for the newly described species of *Kapentagyrus* infecting representatives of *Odaxothrissa* and *Nannothrissa*. The measurements are presented as range followed by average and number of structures measured (n) in parentheses.

Parameter	*K. voreli* n. sp. ex *O. ansorgii*	*K. marispastoris* n. sp. ex *O. mento*	*K. sefcae* n. sp. ex *O. losera*	*K. parisellei* n. sp. ex *N. parva*
**Dorsal anchor**				
Total length	35.6–38.9 (37.5, n = 6)	38.1 (n = 1)	36.7–42 (39.0, n = 10)	18.0–22.2 (20.0, n = 10)
Length to notch	25.3–28.9 (27.8, n = 6)	28.1 (n = 1)	28–31.4 (29.3, n = 10)	13.5–18.3 (16.4, n = 10)
Inner root length	14.9–17.9 (16.0, n = 5)	22.1 (n = 1)	19.8–23.3 (21.6, n = 10)	10.4–13.5 (11.8, n = 9)
Outer root length	4.1–6.4 (4.9, n = 6)	8.8 (n = 1)	5–7.7 (6.2, n = 10)	4.2–7.5 (6.0, n = 9)
Proportion inner/outer root length	2.6–3.8 (3.2, n = 5)	2.5 (n = 1)	2.6–4.2 (3.5, n = 10)	1.4–2.7 (2.0, n = 9)
Proportion inner root length/length of hook pair I	0.9–1.2 (1.1, n = 4)	1.3 (n = 1)	1.2–1.7 (1.4, n = 9)	0.7–0.9 (0.8, n = 6)
Point length	7.8–10.3 (9.4, n = 5)	8.8 (n = 1)	9.6–15.3 (12.3, n = 10)	3.6–7.8 (6.0, n = 8)
**Ventral anchor**				
Total length	35.2–37.3 (36.1, n = 6)	38.2 (n = 1)	41.9–50 (45.1, n = 10)	22.2–26.2 (24.3, n = 10)
Length to notch	26.9–27.8 (27.3, n = 6)	25.5 (n = 1)	18.2–33.3 (30.0, n = 10)	14.2–18.4 (16.9, n = 9)
Inner root length	17–19.2 (18.2, n = 6)	19.0 (n = 1)	22.7–28.7 (26.0, n = 10)	16.4–18.4 (17.0, n = 8)
Outer root length	4.2–6.8 (5.6, n = 5)	10.1 (n = 1)	6.2–10.5 (8.1, n = 9)	3.7–7.5 (5.4, n = 9)
Proportion inner/outer root length	2.8–4.5 (3.4, n = 5)	1.9 (n = 1)	2.6–3.9 (3.3, n = 9)	2.4–4.5 (3.3, n = 8)
Proportion inner root length/length of hook pair I	1.1–1.3 (1.2, n = 5)	1.1 (n = 1)	1.4–1.9 (1.7, n = 9)	1.1–1.4 (1.2, n = 6)
Point length	9.3–11.1 (9.8, n = 6)	10.2 (n = 1)	10.7–15.6 (12.8, n = 10)	5.4–9.4 (7.6, n = 8)
**Dorsal bar**				
Branch length	19.8–23.9 (21.5, n = 6)	27.2 (n = 1)	27.7–31.5 (29.3, n = 10)	19.0–23.0 (21.0, n = 10)
Branch maximum width	5.9–7.3 (6.6, n = 6)	4.9 (n = 1)	7.0–10.4 (8.5, n = 10)	3.5–6 (4.7, n = 10)
**Ventral bar**				
Branch length	23.1–29.7 (25.4, n = 6)	27.4 (n = 1)	25.7–30.9 (27.4, n = 10)	16.2–23.2 (19.0, n = 9)
Branch maximum width	4.7–10.1 (7.7, n = 6)	7.2 (n = 1)	6.1–9.8 (8.5, n = 10)	3.6–5.8 (4.9, n = 9)
**Hooks**				
Average length	16.3–17.8 (17.2, n = 33)	17.5 (n = 5)	13.6–18.5 (17.1, n = 42)	12.4–14.9 (14.1, n = 42)
Pair I	14.2–16.8 (15.6, n = 5)	17.4 (n = 1)	11.9–18.6 (15.7, n = 9)	12.0–15.3 (13.7, n = 8)
Pair II	15.0–20.3 (17.6, n = 5)	17.2 (n = 1)	13.8–18.8 (17, n = 8)	12.4–15.3 (13.8, n = 7)
Pair III	18.1–20.0 (18.8, n = 5)	18.9 (n = 1)	14.5–19.2 (17.0, n = 5)	12.8–15.0 (14.0, n = 6)
Pair IV	15.7–20.8 (18.7, n = 5)	20.3 (n = 1)	14.1–18.2 (16.4, n = 6)	13.1–15.9 (14.4, n = 7)
Pair V	13.6–18.9 (16.7, n = 5)	16.4 (n = 1)	15.8–18.5 (17.8, n = 5)	12.6–13.8 (13.4, n = 3)
Pair VI	13.6–18.8 (16.9, n = 4)	17.1 (n = 1)	17.0–20.0 (18.0, n = 5)	10.8–16.3 (14.2, n = 7)
Pair VII	13.9–16.9 (15.7, n = 4)	15 (n = 1)	17.2–19.7 (18.4, n = 4)	13.2–16.7 (15.3, n = 4)
**Male copulatory organ**				
Copulatory tube axial length	29.0–40.8 (34.4, n = 6)	42.2 (n = 1)	35.5–44.5 (40.6, n = 10)	28.2–36.9 (31.7, n = 11)
Accessory piece axial length	35.1–42.6 (38.1, n = 7)	45.4 (n = 1)	37.8–48.5 (44.8, n = 10)	31.8–45.4 (36.8, n = 11)

**Table 3 animals-11-03578-t003:** Morphometric data (µm) for species of *Kapentagyrus* infecting *Pellonula leonensis*; data for *K. pellonulae* from [32]. The measurements are presented as range followed by average and number of structures measured (n) in parentheses.

Parameter	*K. pellonulae* ex *P. leonensis* (Lake Volta)	*K. hugei* n. sp. ex *P. leonensis* (Lake Volta)	*K. hahni* n. sp. ex *P. leonensis* (Lower Volta)	*K. verbisti* n. sp. ex *P. leonensis* (Gabon)
**Dorsal anchor**				
Total length	40–50	26.0–30.9 (28.2, n = 23)		36.7–38.4 (37.6, n = 2)
Length to notch		21.3–25.1 (22.9, n = 23)		26.4–27.9 (27.2, n = 2)
Inner root length	12	6.7–13.9 (11.2, n = 23)		12.8–17.5 (15.2, n = 2)
Outer root length	1–2	3.3–6.6 (4.5, n = 22)		4–7 (5.5, n = 2)
Proportion inner/outer root length		1.6–3.6 (2.5, n = 21)		2.5–3.2 (2.9, n = 2)
Proportion inner root length/length of hook pair I		0.4–0.9 (0.7, n = 17)		0.8 (n = 1)
Point length		7.0–10.8 (9.0, n = 23)		9.7–11.7 (10.7, n = 2)
**Ventral anchor**				
Total length	50	24.3–31.6 (27.7, n = 30)	24.1–29.3 (26.2, n = 3)	37.8 (n = 1)
Length to notch		19.4–25.0 (21.7, n = 30)	13.1–18.8 (15.4, n = 3)	28.5 (n = 1)
Inner root length	15–18	11.4–18.9 (15.1, n = 30)	14.3–17.5 (15.0, n = 3)	21.9 (n = 1)
Outer root length	4–5	3.4–6.0 (4.7, n = 23)	4.2–4.7 (4.5, n = 2)	5.2 (n = 1)
Proportion inner/outer root length		2.6–4.3 (3.4, n = 23)	3.4–3.7 (3.6, n = 2)	4.2 (n = 1)
Proportion inner root length/length of hook pair I		0.8–1.2 (1.0, n = 22)		
Point length		7.1–9.4 (8.4, n = 29)	6.8–9.9 (7.9, n = 3)	10.4 (n = 1)
**Dorsal bar**				
Branch length	30–40	17.4–22.4 (20.3, n = 22)	18.5 (18.5, n = 2)	25.9–27.3 (26.6, n = 2)
Branch maximum width		3.3–6 (4.6, n = 20)		6.6–6.9 (6.8, n = 2)
**Ventral bar**				
Branch length	25–35	12.2–22.9 (19.5, n = 30)	18.0–20.4 (18.9, n = 3)	
Branch maximum width		4.1–7.8 (5.5, n = 30)	5–5.9 (5.5, n = 3)	8.9–9.4 (9.2, n = 2)
**Hooks**				
Average length	8–15	13.5–17.5 (16.2, n = 127)	13.6–15.7 (14.6, n = 8)	18.1–20.4 (19.5, n = 16)
Pair I		12.1–16.5 (15.1, n = 23)		17.0–19,1 (18.1, n = 2)
Pair II		13.4–17.4 (15.8, n = 18)	13.4–17.0 (14.8, n = 3)	16.7–22.4 (20.0, n = 3)
Pair III		14–19.3 (16.9, n = 22)	13.2–16.0 (14.6, n = 2)	17.7–18.9 (18.3, n = 2)
Pair IV		14.9–19.7 (17.2, n = 20)	16.8 (n = 1)	20.2–23.4 (21.8, n = 2)
Pair V		11.1–17.5 (15.6, n = 14)		16.6–20.2 (18.4, n = 3)
Pair VI		14.7–18.8 (16.8, n = 17)	11.8–14.4 (13.1, n = 2)	18.7–19.1 (18.9, n = 2)
Pair VII		15–18.9 (16.9, n = 13)		19.3–19.6 (19.5, n = 2)
**Male copulatory organ**				
Copulatory tube axial length	25	23.5–31.4 (27.9, n = 22)		35.2–37.2 (36.2, n = 2)
Accessory piece axial length	30	22.4–34.0 (28.8, n = 22)		36.6–47.8 (42.2, n = 2)

**Table 4 animals-11-03578-t004:** Morphometric data (µm) for species of *Kapentagyrus* infecting members of *Microthrissa* or *Potamothrissa*. The measurements are presented as range followed by average and number of structures measured (n) in parentheses.

Parameter	*K. chochamandai* n. sp. ex *M. moeruensis*	*K. bisthoveni* n. sp. ex *M. congica*	*K. boegeri* n. sp. ex *M. congica*	*K. rochetteae* n. sp. ex *M. royauxi*	*K. chochamandai* n. sp. ex *P. acutirostris*
**Dorsal anchor**					
Total length	22.0–25.5 (23.7, n = 7)	20.4–26.7 (24.8, n = 4)	22.5–29.6 (26.1, n = 2)	20.3–27.8 (24.2, n = 9)	22.7 (n = 1)
Length to notch	16.8–19.4 (18.0, n = 7)	14.7–18.5 (17.2, n = 4)	16.0–24.1 (20.0, n = 2)	15.2–20.3 (17.3, n = 9)	18.0 (n = 1)
Inner root length	11.1–14.9 (12.7, n = 7)	13.4–18.5 (16.3, n = 4)	11.0–11.5 (11.2, n = 2)	13.6–18.1 (15.4, n = 9)	11.0 (n = 1)
Outer root length	5.2–7.0 (6.2, n = 7)	3.5–7.0 (5.4, n = 4)	3.6–6.1 (4.8, n = 2)	5.1–7.8 (6.3, n = 9)	5.5 (n = 1)
Proportion inner/outer root length	1.8–2.5 (2.1, n = 7)	2.3–3.8 (3.1, n = 4)	1.8–3.2 (2.5, n = 2)	2.1–2.9 (2.5, n = 9)	2 (n = 1)
Proportion inner root length/length of hook pair I	0.7–0.9 (0.8, n = 6)	0.9–1.5 (1.2, n = 4)		1.1–1.6 (1.3, n = 8)	
Point length	5.0–6.4 (5.7, n = 7)	4.0–5.9 (5.2, n = 3)	7.3–8.0 (7.7, n = 2)	5.3–7.5 (6.8, n = 9)	
**Ventral anchor**					
Total length	27.9–33.6 (29.6, n = 6)	25.6–27.6 (26.6, n = 3)	23.3–25.5 (24.3, n = 3)	25.1–29.9 (27.7, n = 9)	23.6 (n = 1)
Length to notch	19.6–22.6 (21.2, n = 7)	14.0–20.6 (17.9, n = 3)	15.1–21.6 (18.6, n = 3)	16.8–19.4 (18.2, n = 9)	18.5 (n = 1)
Inner root length	13.8–17.5 (16.3, n = 7)	17.1–19.6 (18.2, n = 3)	16.2–16.8 (16.4, n = 3)	17.0–21.2 (19.3, n = 9)	14.5 (n = 1)
Outer root length	5.7–8.4 (7.1, n = 7)	7.3–7.4 (7.4, n = 2)	4.6–7.1 (6.2, n = 3)	4.7–8.4 (6.9, n = 9)	6.1 (n = 1)
Proportion inner/outer root length	1.9–3.0 (2.3, n = 7)	2.3–2.7 (2.5, n = 2)	2.3–3.5 (2.8, n = 3)	2.2–4.5 (3.0, n = 9)	2.4 (n = 1)
Proportion inner root length/length of hook pair I	0.9–1.2 (1.1, n = 6)	1.1–1.6 (1.4, n = 3)		1.3–1.8 (1.6, n = 8)	
Point length	5.4–9.2 (7.3, n = 7)	5.5–9.0 (7.6, n = 3)	6.0–7.3 (6.6, n = 2)	6.4–8.9 (7.4, n = 9)	7.2 (n = 1)
**Dorsal bar**					
Branch length	17.7–23.9 (20.1, n = 7)	16.8–20.9 (19.4, n = 4)	23.0–25.0 (23.8, n = 3)	16.3–23.5 (20.6, n = 9)	16.1 (n = 1)
Branch maximum width	3.8–6.6 (5.1, n = 7)	5.2–6.7 (5.8, n = 4)	3.9–5.8 (4.8, n = 2)	3.7–6.0 (5.1, n = 9)	5.0 (n = 1)
**Ventral bar**					
Branch length	17.2–21.8 (20.3, n = 6)	20.2–22.1 (21.3, n = 3)	21.0–22.2 (21.6, n = 2)	15.2–21.3 (18.3, n = 9)	
Branch maximum width	4.5–8.9 (5.8, n = 7)	5.2–6.7 (5.8, n = 4)	3.4–5.5 (4.7, n = 3)	4.3–6.5 (5.4, n = 9)	
**Hooks**					
Average length	14.1–16.9 (15.8, n = 35)	14.3–16.9 (15.2, n = 16)	13.6–16.8 (15.3, n = 12)	9.7–13.3 (12.1, n = 47)	17.7 (n = 4) (excluding pair V) 17.1 (n = 5) (including pair V)
Pair I	13.5–18.0 (15.6, n = 6)	12.0–15.8 (13.7; n = 4)	16.2 (n = 1)	11.1–13.9 (12.2, n = 8)	
Pair II	14.8–17.3 (16.1, n = 5)	13.1–16.8 (14.9, n = 2)	14.4–17.9 (16.1, n = 2)	9.3–15.4 (12.2, n = 9)	
Pair III	13.5–17.8 (15.9, n = 5)	15.3–18.1 (16.7, n = 2)	13.3–16.5 (15.0, n = 3)	6.9–14.1 (11.6, n = 8)	
Pair IV	13.4–17.6 (15.9, n = 5)	15.3–18.0 (16.6, n = 2)	13.8–15.9 (14.8, n = 2)	9.4–14.3 (11.7, n = 8)	
Pair V	13.1–16.1 (14.5, n = 5)	16.0–16.1 (16.0, n = 2)	13.7–16.5 (15.1, n = 2)	10.0–12.1 (11.2, n = 5)	14.6 (n = 1)
Pair VI	14.0–20.2 (16.5, n = 5)	15.5–16.6 (16.0, n = 2)	16.9 (n = 1)	12.6–14.4 (13.3, n = 5)	
Pair VII	15.3–22.5 (17.6, n = 4)	15.1–16.6 (15.8, n = 2)	16.1 (n = 1)	13.0–15.1 (13.7, n = 4)	
**Male copulatory organ**					
Copulatory tube axial length	25.0–33.0 (28.5, n = 7)	23.0–36.2 (31.8, n = 4)	20.6–41.3 (28.8, n = 3)	27.4–38.3 (32.7, n = 11)	32.7 (n = 1)
Accessory piece axial length	28.5–34.9 (32.2, n = 6)	26.6–43.2 (34.9, n = 4)	22.8–41.9 (32.4, n = 2)	32.8–38.7 (36.1, n = 11)	35.4 (n = 1)

## Data Availability

Type material was deposited in the invertebrate collection of the RMCA (RMCA_VERMES_43430-459), the collection of the research group Zoology: Biodiversity and Toxicology of Hasselt University (Diepenbeek, Belgium) (HU 779-820), the Finnish Museum of Natural History (Helsinki, Finland) (MZH https://id.luomus.fi/KV.670-https://id.luomus.fi/KV.685), and the Iziko South African Museum (Cape Town, South Africa) (SAMC-A094487-504). Raw morphometric data are available in Appendix A.

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
