# Peer review of "From the Atlantic Coast to Lake Tanganyika: Gill-Infecting Flatworms of Freshwater Pellonuline Clupeid Fishes in West and Central Africa, with Description of Eleven New Species and Key to Kapentagyrus (Monogenea, Dactylogyridae)"

_animals, 2021, doi:10.3390/ani11123578_

Round 1

Reviewer 1 Report

The manuscript presents important register of new species of monogenean apart from the low number of monogeneans collected in some fish species. Figures present high quality and tables present the data in detail. I have some few suggestions to improve the understanding of the reader.

Tables: Morphometric data (µm) for….: include (µm)

Include in legend: The measurements are presented as variation amplitude followed by means and number of structures measured (n) in parenthesis.

Results: after each host in taxonomic summary, include the host total length.

It needs some legend for Supplementary material. In the excel file please complete which means the letters of structures measurements.

Author Response

The manuscript presents important register of new species of monogenean apart from the low number of monogeneans collected in some fish species. Figures present high quality and tables present the data in detail. I have some few suggestions to improve the understanding of the reader.

RESPONSE: many thanks for your positive appraisal of our manuscript!

Tables: Morphometric data (µm) for….: include (µm)

RESPONSE: this was now added in the caption of Tables 2, 3 and 4.

Include in legend: The measurements are presented as variation amplitude followed by means and number of structures measured (n) in parenthesis.

RESPONSE: we added the following in the caption of Tables 2, 3 and 4: “The measurements are presented as range followed by average and number of structures measured (n) in parentheses.”

Results: after each host in taxonomic summary, include the host total length.

RESPONSE: thanks for this suggestion. Unfortunately, we did not measure this on all host specimens used for this study. Given these specimens are held in two enormous institutional collections in Belgium (Royal Museum for Central Africa) and abroad (Natural History Museum), we will not be able to retrieve and measure them within the deadline communicated to us for resubmission of this manuscript. Moreover, given the COVID-19 measures in place, it is unlikely these collections will be swiftly accessible to us in the near future.

It needs some legend for Supplementary material. In the excel file please complete which means the letters of structures measurements.

RESPONSE: we now added a legend in a footnote (tab about the haptor) and spelled the metrics out in full in the title row (tab about the MCO) in the supplementary table.

Reviewer 2 Report

Page 3, line 103: As Pellonulini … I would suggest to add the term tribus. As the tribus Pellonulini …
Figure 1 is very difficult to read and should be modified accordingly.
Page 39, line 719: scrutiny should be replaced by examination.

Author Response

Page 3, line 103: As Pellonulini … I would suggest to add the term tribus. As the tribus Pellonulini …

RESPONSE: we have added the anglicized version of this term, “tribe”.

Figure 1 is very difficult to read and should be modified accordingly.

RESPONSE: we have inserted a legend inside Fig. 1 to clarify things.

Page 39, line 719: scrutiny should be replaced by examination.

RESPONSE: we changed this accordingly.

Reviewer 3 Report

Authors use samples from Natural History museums to screen for new species of monogenea parasites in clupeid fishes in different regions of Africa. The study presents interest as they describe new 11 species of flatworms and by using samples that belong to underestudied regions.

Nevertheless, there are some changes that could lead to a better quality of the manuscript :

-The map is very useful in order to indicate the procedure of the samples but there are many symbols and it becomes a bit messy or difficult to underestand. The inclusion of a caption is necesary.

-In a similar way, drawings and pictures are amazing but some schematic protocol would be helpful to indicate all the measurements performed. Perhaps, by using one of the drawings and indicating with arrows and the name of each meassurement next to it.

-Also, it would be helpful  to summarise the data in a final table comparing the features of the species described.

-Information regarding the status of the samples is missing. How are this samples preserved?

Regarding a different aspect, while hook measurements, etc. are considered taxonomic features for Monogenea, nowadays, it is very relevant to look as well into genetic features in order to describe new species. I highly recommend to try to extract DNA from the samples and make some sequencing analysis in order to complete and support the conclusions of the study. By doing this, euthors would be also able to perform some phylogenetic analysis that could show the relationship of the different species found and study it respect to their original location which will highly improve the quality of the research.

Author Response

Authors use samples from Natural History museums to screen for new species of monogenea parasites in clupeid fishes in different regions of Africa. The study presents interest as they describe new 11 species of flatworms and by using samples that belong to underestudied regions.

RESPONSE: many thanks for your interest in our study!

Nevertheless, there are some changes that could lead to a better quality of the manuscript :

-The map is very useful in order to indicate the procedure of the samples but there are many symbols and it becomes a bit messy or difficult to underestand. The inclusion of a caption is necesary.

RESPONSE: we followed this suggestion regarding Fig. 1.

-In a similar way, drawings and pictures are amazing but some schematic protocol would be helpful to indicate all the measurements performed. Perhaps, by using one of the drawings and indicating with arrows and the name of each meassurement next to it.

RESPONSE: thanks for this suggestion. We have now added an additional figure (Fig. 2), and refer to it in the Materials and Methods section.

-Also, it would be helpful  to summarise the data in a final table comparing the features of the species described.

RESPONSE: we prefer not to comply with this suggestion, as we think this would be redundant and hence render the manuscript unnecessarily lengthy, because (1) Tables 2, 3, and 4 already provide a comparison of measurements, and a similar table would never allow the inclusion of all species discussed in this manuscript, and (2) for the diagnostic features allowing species identification, the dichotomous key in section 3.4 of the manuscript provides easy access to species traits.

-Information regarding the status of the samples is missing. How are this samples preserved?

RESPONSE: we added this information on lines 114-115.

Regarding a different aspect, while hook measurements, etc. are considered taxonomic features for Monogenea, nowadays, it is very relevant to look as well into genetic features in order to describe new species. I highly recommend to try to extract DNA from the samples and make some sequencing analysis in order to complete and support the conclusions of the study. By doing this, euthors would be also able to perform some phylogenetic analysis that could show the relationship of the different species found and study it respect to their original location which will highly improve the quality of the research.

RESPONSE: we naturally agree with this constructive suggestion of the reviewer and hope to one day be able to put it into practice. We even took this strategy into account in our sampling protocol, as was explained in the text (lines 124-125, lines 871-873). However, at the current moment this is impossible with the material available to us, and we explain this limitation already in the original manuscript (lines 864-868).